# Stabilizing Reinforcement Learning in Differentiable Multiphysics Simulation

**Eliot Xing & Vernon Luk & Jean Oh**
Carnegie Mellon University
`{etaoxing, vluk, jeanoh}@cmu.edu`

## Abstract

Recent advances in GPU-based parallel simulation have enabled practitioners to collect large amounts of data and train complex control policies using deep reinforcement learning (RL), on commodity GPUs. However, such successes for RL in robotics have been limited to tasks sufficiently simulated by fast rigid-body dynamics. Simulation techniques for soft bodies are comparatively several orders of magnitude slower, thereby limiting the use of RL due to sample complexity requirements. To address this challenge, this paper presents both a novel RL algorithm and a simulation platform to enable scaling RL on tasks involving rigid bodies and deformables. We introduce Soft Analytic Policy Optimization (SAPO), a maximum entropy first-order model-based actor-critic RL algorithm, which uses first-order analytic gradients from differentiable simulation to train a stochastic actor to maximize expected return and entropy. Alongside our approach, we develop Rewarped, a parallel differentiable multiphysics simulation platform that supports simulating various materials beyond rigid bodies. We re-implement challenging manipulation and locomotion tasks in Rewarped, and show that SAPO outperforms baselines over a range of tasks that involve interaction between rigid bodies, articulations, and deformables. Additional details at `rewarped.github.io`.

## 1 Introduction

Progress in deep reinforcement learning (RL) has produced policies capable of impressive behavior, from playing games with superhuman performance (Silver et al., 2016; Vinyals et al., 2019) to controlling robots for assembly (Tang et al., 2023), dexterous manipulation (Andrychowicz et al., 2020; Akkaya et al., 2019), navigation (Wijmans et al., 2020; Kaufmann et al., 2023), and locomotion (Rudin et al., 2021; Radosavovic et al., 2024). However, standard model-free RL algorithms are extremely sample inefficient. Thus, the main practical bottleneck when using RL is the cost of acquiring large amounts of training data.

To scale data collection for online RL, prior works developed distributed RL frameworks (Nair et al., 2015; Horgan et al., 2018; Espeholt et al., 2018) that run many processes across a large compute cluster, which is inaccessible to most researchers and practitioners. More recently, GPU-based parallel environments (Dalton et al., 2020; Freeman et al., 2021; Liang et al., 2018; Makoviychuk et al., 2021; Mittal et al., 2023; Gu et al., 2023) have enabled training RL at scale on a single consumer GPU.

However, such successes of scaling RL in robotics have been limited to tasks sufficiently simulated by fast rigid-body dynamics (Makoviychuk et al., 2021), while physics-based simulation techniques for soft bodies are comparatively several orders of magnitude slower. Consequently for tasks involving deformable objects, such as robotic manipulation of rope (Nair et al., 2017; Chi et al., 2022), cloth (Ha & Song, 2022; Lin et al., 2022), elastics (Shen et al., 2022), liquids (Ichnowski et al., 2022; Zhou et al., 2023), dough (Shi et al., 2022; 2023; Lin et al., 2023), or granular piles (Wang et al., 2023; Xue et al., 2023), approaches based on motion planning, trajectory optimization, or model predictive control have been preferred over and outperform RL (Huang et al., 2020; Chen et al., 2022).

How can we overcome this data bottleneck to scaling RL on tasks involving deformables? Model-based reinforcement learning (MBRL) has shown promise at reducing sample complexity, by leveraging some known model or learning a world model to predict environment dynamics and rewards (Moerland et al., 2023). In contrast to rigid bodies however, soft bodies have more complex dynamics

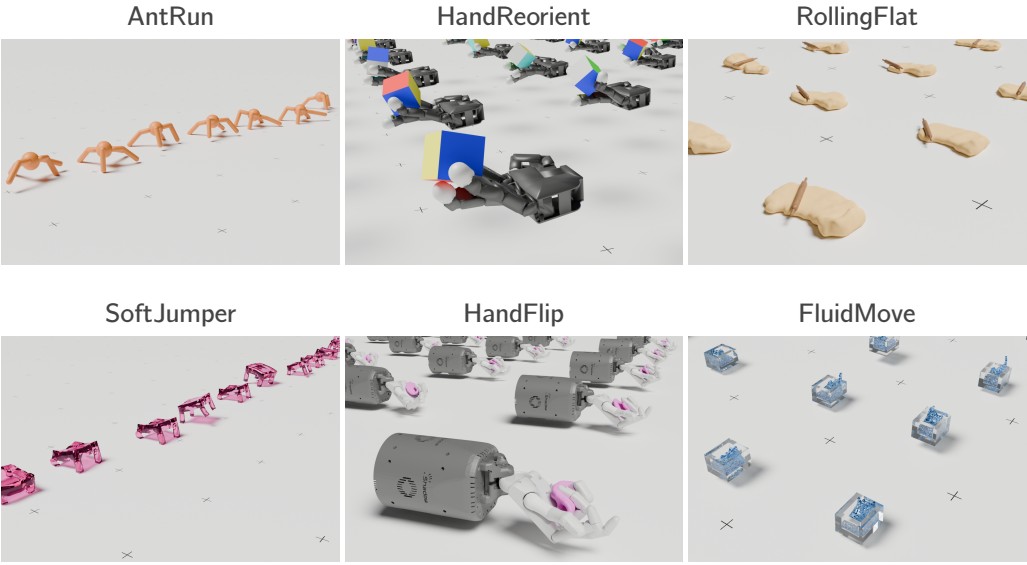

*Figure 1:* **Visualizations of tasks implemented in Rewarped.** These are manipulation and locomotion tasks involving rigid and soft bodies. AntRun and HandReorient are tasks with articulated rigid bodies, while RollingFlat, SoftJumper, HandFlip, and FluidMove are tasks with deformables.

and higher-dimensional state spaces. This makes learning to model dynamics of deformables highly nontrivial (Lin et al., 2021), often requiring specialized systems architecture and material-specific assumptions such as volume preservation or connectivity.

Recent developments in differentiable physics-based simulators of deformables (Hu et al., 2019b; Du et al., 2021; Huang et al., 2020; Zhou et al., 2023; Wang et al., 2024; Liang et al., 2019; Qiao et al., 2021a; Li et al., 2022b; Heiden et al., 2023) have shown that first-order gradients from differentiable simulation can be used for gradient-based trajectory optimization and achieve low sample complexity. Yet such approaches are sensitive to initial conditions and get stuck in local optima due to non-smooth optimization landscapes or discontinuities induced by contacts (Li et al., 2022a; Antonova et al., 2023). Additionally, existing soft-body simulations are not easily parallelized, which limits scaling RL in them. Overall, there is no existing simulation platform that is parallelized, differentiable, and supports interaction between articulated rigid bodies and deformables.

In this paper, we approach the sample efficiency problem using first-order model-based RL (FO-MBRL), which leverages first-order analytic gradients from differentiable simulation to accelerate policy learning, without explicitly learning a world model. Thus far, FO-MBRL has been shown to achieve low sample complexity on articulated rigid-body locomotion tasks (Freeman et al., 2021; Xu et al., 2021), but has not yet been shown to work well for tasks involving deformables (Chen et al., 2022). We hypothesize that entropy regularization can stabilize policy optimization over analytic gradients from differentiable simulation, such as by smoothing the optimization landscape (Ahmed et al., 2019). To this end, we introduce a novel maximum entropy FO-MBRL algorithm, alongside a parallel differentiable multiphysics simulation platform for RL.

**Contributions. i)** We introduce Soft Analytic Policy Optimization (SAPO), a first-order MBRL algorithm based on the maximum entropy RL framework. We formulate SAPO as an on-policy actor-critic RL algorithm, where a stochastic actor is trained to maximize expected return and entropy using first-order analytic gradients from differentiable simulation. **ii)** We present Rewarped, a scalable and easy-to-use platform which enables parallelizing RL environments of GPU-accelerated differentiable multiphysics simulation and supports various materials beyond rigid bodies. **iii)** We demonstrate that parallel differentiable simulation enables SAPO to outperform baselines over a range of challenging manipulation and locomotion tasks re-implemented using Rewarped that involve interaction between rigid bodies, articulations, and deformables such as elastic, plasticine, or fluid materials.

## 2 RELATED WORK

We refer the reader to (Newbury et al., 2024) for an overview of differentiable simulation. We cover *non-parallel* differentiable simulation and model-based RL in Appendix A.

| Simulator | $\nabla$? | Materials? | | | | |
|---|---|---|---|---|---|---|
| | | Rigid | Articulated | Elastic | Plasticine | Fluid |
| Isaac Gym | ✗ | ✓ | ✓ | ✓ | ✗ | ✗ |
| Isaac Lab / Orbit | ✗ | ✓ | ✓ | ✓* | ✗ | ✓* |
| ManiSkill | ✗ | ✓ | ✓ | ✗ | ✓* | ✓* |
| TinyDiffSim | ✓ | ✓ | ✓ | ✗ | ✗ | ✗ |
| Brax | ✓ | ✓ | ✓ | ✗ | ✗ | ✗ |
| MJX | ✓* | ✓ | ✓ | ✗ | ✗ | ✗ |
| DaXBench | ✓ | ✓ | ✗ | ✗ | ✓* | ✓ |
| DFlex | ✓ | ✓ | ✓ | ✓* | ✗ | ✗ |
| Rewarped (ours) | ✓ | ✓ | ✓ | ✓ | ✓ | ✓ |

*Table 1:* **Comparison of physics-based parallel simulation platforms for RL.** We use * to indicate incomplete feature support at the time of writing. i) Isaac Lab / Orbit: Missing deformable tasks due to breaking API changes and poor simulation stability / scaling. ii) ManiSkill: The latest version ManiSkill3 does not yet support the soft body tasks introduced in v2. iii) MJX: Stability issues with autodifferentiation and gradients. iv) DaXBench: Plasticine task was omitted from benchmark and requires additional development. v) DFlex: While later work (Murthy et al., 2021; Heiden et al., 2023) has built on DFlex to support elastic and cloth materials, their simulations were not parallelized.

**Parallel differentiable simulation.** There are few prior works on parallel differentiable simulators capable of running many environments together, while also computing simulation gradients in batches. TinyDiffSim (Heiden et al., 2021) implements articulated rigid-body dynamics and contact models in C++/CUDA that can integrate with various autodifferentiation libraries. Brax (Freeman et al., 2021) implements a parallel simulator in JAX for articulated rigid-body dynamics with simple collision shape primitives. Recently, MJX is building on Brax to provide a JAX re-implementation of MuJoCo (Todorov et al., 2012), a physics engine widely used in RL and robotics, but does not have feature parity with MuJoCo yet. These aforementioned parallel differentiable simulators are only capable of modeling articulated rigid bodies. DaXBench (Chen et al., 2022) also uses JAX to enable fast parallel simulation of deformables such as rope and liquid by Material Point Method (MPM) or cloth by mass-spring systems, but does not support articulated rigid bodies. DFlex (Xu et al., 2021) presents a differentiable simulator based on source-code transformation (Griewank & Walther, 2008; Hu et al., 2020) of simulation kernel code to C++/CUDA, that integrates with PyTorch for tape-based autodifferentiation. Xu et al. (2021) use DFlex for parallel simulation of articulated rigid bodies for high-dimensional locomotion tasks. Later work (Murthy et al., 2021; Heiden et al., 2023) also used DFlex to develop differentiable simulations of cloth and elastic objects, but these were not parallelized and did not support interaction with articulated rigid bodies. To the best of our knowledge, there is no existing differentiable simulation platform that is parallelized with multiphysics support for interaction between rigid bodies, articulations, and various deformables. In this paper, we aim to close this gap with Rewarped, our platform for parallel differentiable multiphysics simulation, and in Table 1 we compare Rewarped against existing physics-based parallel simulation platforms.

**Learning control with differentiable physics.** Gradient-based trajectory optimization is commonly used with differentiable simulation of soft bodies (Hu et al., 2019b; 2020; Huang et al., 2020; Li et al., 2023a; Zhou et al., 2023; Wang et al., 2024; Si et al., 2024; Du et al., 2021; Li et al., 2022b; Rojas et al., 2021; Qiao et al., 2020; 2021a; Liu et al., 2024; Chen et al., 2022; Heiden et al., 2023). Differentiable physics can provide physical priors for control in end-to-end learning systems, such as for quadruped locomotion (Song et al., 2024), drone navigation (Zhang et al., 2024), robot painting (Schaldenbrand et al., 2023), or motion imitation (Ren et al., 2023). Gradients from differentiable simulation can also be directly used for policy optimization. PODS (Zamora et al., 2021) proposes a first and second order policy improvement based on analytic gradients of a value function with respect to the policy's action outputs. APG (Freeman et al., 2021) uses analytic simulation gradients to directly compute policy gradients. SHAC (Xu et al., 2021) presents an actor-critic algorithm, where the actor is optimized over a short horizon using analytic gradients, and a terminal value function helps smooth the optimization landscape. AHAC (Georgiev et al., 2024) modifies SHAC to adjust the policy horizon by truncating stiff contacts based on contact forces or the norm of the dynamics Jacobian. Several works also propose different ways to overcome bias and non-smooth dynamics resulting from contacts, by reweighting analytic gradients (Gao et al., 2024; Son et al., 2024) or

explicit smoothing (Suh et al., 2022; Zhang et al., 2023; Schwarke et al., 2024). In this work, we propose a maximum entropy FO-MBRL algorithm to stabilize policy learning with gradients from differentiable simulation.

## 3 BACKGROUND

**Reinforcement learning** (RL) considers an agent interacting with an environment, formalized as a Markov decision process (MDP) represented by a tuple $(\mathcal{S}, \mathcal{A}, P, R, \rho_0, \gamma)$. In this work, we consider discrete-time, infinite-horizon MDPs with continuous action spaces, where $\boldsymbol{s} \in \mathcal{S}$ are states, $\boldsymbol{a} \in \mathcal{A}$ are actions, $P : \mathcal{S} \times \mathcal{A} \to \mathcal{S}$ is the transition function, $R : \mathcal{S} \times \mathcal{A} \to \mathbb{R}$ is a reward function, $\rho_0(\boldsymbol{s})$ is an initial state distribution, and $\gamma$ is the discount factor. We want to obtain a policy $\pi : \mathcal{S} \to \mathcal{A}$ which maximizes the expected discounted sum of rewards (return) $\mathbb{E}_\pi[\sum_{t=0}^{\infty} \gamma^t r_t]$ with $r_t = R(\boldsymbol{s}_t, \boldsymbol{a}_t)$, starting from state $\boldsymbol{s}_0 \sim \rho_0$. We also denote the state distribution $\rho_\pi(\boldsymbol{s})$ and state-action distribution $\rho_\pi(\boldsymbol{s}, \boldsymbol{a})$ for trajectories generated by a policy $\pi(\boldsymbol{a}_t|\boldsymbol{s}_t)$.

In practice, the agent interacts with the environment for $T$ steps in a finite-length episode, yielding a trajectory $\tau = (\boldsymbol{s}_0, \boldsymbol{a}_0, \boldsymbol{s}_1, \boldsymbol{a}_1, \dots, \boldsymbol{s}_{T-1}, \boldsymbol{a}_{T-1})$. We can define the $H$-step return :

$$R_{0:H}(\tau) = \sum_{t=0}^{H-1} \gamma^t r_t, \tag{1}$$

and standard RL objective to optimize $\theta$ parameterizing a policy $\pi_\theta$ to maximize the expected return :

$$J(\pi) = \mathbb{E}_{\substack{\boldsymbol{s}_0 \sim \rho_0 \\ \tau \sim \rho_\pi}}[R_{0:T}]. \tag{2}$$

Typically, the policy gradient theorem (Sutton et al., 1999) provides a useful expression of $\nabla_\theta J(\pi)$ that does not depend on the derivative of state distribution $\rho_\pi(\cdot)$ :

$$\nabla_\theta J(\pi) \propto \int_{\mathcal{S}} \rho_\pi(\boldsymbol{s}) \int_{\mathcal{A}} \nabla_\theta \pi(\boldsymbol{a}|\boldsymbol{s}) Q^\pi(\boldsymbol{s}, \boldsymbol{a}) \, d\boldsymbol{a} \, d\boldsymbol{s}, \tag{3}$$

where $Q^\pi(\boldsymbol{s}_t, \boldsymbol{a}_t) = \mathbb{E}_{\tau \sim \rho_\pi}[R_{t:T}]$ is the $Q$-function (state-action value function).

We proceed to review zeroth-order versus first-order estimators of the policy gradient following the discussion in (Suh et al., 2022; Georgiev et al., 2024). We denote a single zeroth-order estimate :

$$\hat{\nabla}_\theta^{[0]} J(\pi) = R_{0:T} \sum_{t=0}^{T-1} \nabla_\theta \log \pi(\boldsymbol{a}_t|\boldsymbol{s}_t), \tag{4}$$

where the zeroth-order batched gradient (ZOBG) is the sample mean $\overline{\nabla}_\theta^{[0]} J(\pi) = \frac{1}{N} \sum_{i=1}^{N} \hat{\nabla}_\theta^{[0]} J(\pi)$ and is an unbiased estimator, under some mild assumptions to ensure the gradients are well-defined. The ZOBG yields an $N$-sample Monte-Carlo estimate commonly known as the REINFORCE estimator (Williams, 1992) in RL literature, or the score function / likelihood-ratio estimator. Policy gradient methods may use different forms of Equation 4 to adjust the bias and variance of the estimator (Schulman et al., 2015b). For instance, a baseline term can be used to reduce variance of the estimator, by substituting $R_{0:T}$ with $R_{0:T} - R_{l:H+l}$.

**Differentiable simulation** as the environment provides gradients for the transition dynamics $P$ and rewards $R$, so we can directly obtain an analytic value for $\nabla_\theta R_{0:T}$ under policy $\pi_\theta$. In this setting, for a single first-order estimate :

$$\hat{\nabla}_\theta^{[1]} J(\pi) = \nabla_\theta R_{0:T}, \tag{5}$$

then the first-order batched gradient (FOBG) is the sample mean $\overline{\nabla}_\theta^{[1]} J(\pi) = \frac{1}{N} \sum_{i=1}^{N} \hat{\nabla}_\theta^{[1]} J(\pi)$, and is also known as the pathwise derivative (Schulman et al., 2015a) or reparameterization trick (Kingma & Welling, 2014; Rezende et al., 2014; Titsias & Lázaro-Gredilla, 2014).

**First-order model-based RL** (FO-MBRL) aims to use differentiable simulation (and its first-order analytic gradients) as a known differentiable model, in contrast to vanilla MBRL which either assumes a given non-differentiable model or learns a world model of dynamics and rewards from data.

**Analytic Policy Gradient** (APG, Freeman et al. (2021)) uses FOBG estimates to directly maximize the discounted return over a truncated horizon :

$$J(\pi) = \sum_{l=t}^{t+H-1} \mathbb{E}_{(\boldsymbol{s}_l, \boldsymbol{a}_l) \sim \rho_\pi}[\gamma^{l-t} r_l], \tag{6}$$

and is also referred to as Backpropagation Through Time (BPTT, Werbos (1990); Mozer (1995)), particularly when the horizon is the full episode length (Degrave et al., 2019; Huang et al., 2020).

**Short-Horizon Actor-Critic** (SHAC, Xu et al. (2021)) is a FO-MBRL algorithm which learns a policy $\pi_\theta$ and (terminal) value function $V_\psi$ :

$$J(\pi) = \sum_{l=t}^{t+H-1} \mathbb{E}_{(\boldsymbol{s}_l, \boldsymbol{a}_l) \sim \rho_\pi}[\gamma^{l-t} r_l + \gamma^t V(\boldsymbol{s}_{t+H})], \tag{7}$$

$$\mathcal{L}(V) = \sum_{l=t}^{t+H-1} \mathbb{E}_{\boldsymbol{s}_l \sim \rho_\pi}[||V(\boldsymbol{s}) - \tilde{V}(\boldsymbol{s})||^2], \tag{8}$$

where $\tilde{V}(\boldsymbol{s}_t)$ are value estimates for state $\boldsymbol{s}_t$ computed starting from time step $t$ over an $H$-step horizon. TD($\lambda$) (Sutton, 1988) is used for value estimation, which computes $\lambda$-returns $G^\lambda_{t:t+H}$ as a weighted average of value-bootstrapped $k$-step returns $G_{t:t+k}$ :

$$\tilde{V}(\boldsymbol{s}_t) = G^\lambda_{t:t+H} = (1-\lambda)\left(\sum_{l=1}^{H-1-t} \lambda^{l-1} G_{t:t+l}\right) + \lambda^{H-t-1} G_{t:t+H}, \tag{9}$$

where $G_{t:t+k} = \left(\sum_{l=0}^{k-1} \gamma^l r_{t+l}\right) + \gamma^k V(\boldsymbol{s}_{t+k})$. The policy and value function are optimized in an alternating fashion per standard actor-critic formulation (Konda & Tsitsiklis, 1999). The policy gradient is obtained by FOBG estimation, with single first-order estimate :

$$\hat{\nabla}^{[1]}_\theta J(\pi) = \nabla_\theta(R_{0:H} + \gamma^H V(\boldsymbol{s}_H)), \tag{10}$$

and the value function is optimized as usual by backpropagating $\nabla_\psi \mathcal{L}(V)$ of the mean-squared loss in Eq. 8. Combining value estimation with a truncated short-horizon window where $H \ll T$ (Williams & Zipser, 1995), SHAC optimizes over a smoother surrogate reward landscape compared to BPTT over the entire $T$-step episode.

## 4  SOFT ANALYTIC POLICY OPTIMIZATION (SAPO)

Empirically we observe that SHAC, a state-of-the-art FO-MBRL algorithm, is still prone to suboptimal convergence to local minima in the reward landscape (Appendix, Figure 5). We hypothesize that entropy regularization can stabilize policy optimization over analytic gradients from differentiable simulation, such as by smoothing the optimization landscape (Ahmed et al., 2019) or providing robustness under perturbations (Eysenbach & Levine, 2022).

We draw on the maximum entropy RL framework (Kappen, 2005; Todorov, 2006; Ziebart et al., 2008; Toussaint, 2009; Theodorou et al., 2010; Haarnoja et al., 2017) to formulate Soft Analytic Policy Optimization (SAPO), a maximum entropy FO-MBRL algorithm (Section 4.1). To implement SAPO, we make several design choices, including modifications building on SHAC (Section 4.2). In Appendix B.1, we describe how we use visual encoders to learn policies from high-dimensional visual observations in differentiable simulation. Pseudocode for SAPO is shown in Appendix B.2, and the computational graph of SAPO is illustrated in Appendix Figure 4.

### 4.1  MAXIMUM ENTROPY RL IN DIFFERENTIABLE SIMULATION

**Maximum entropy RL** (Ziebart et al., 2008; Ziebart, 2010) augments the standard (undiscounted) return maximization objective with the expected entropy of the policy over $\rho_\pi(\boldsymbol{s}_t)$ :

$$J(\pi) = \sum_{t=0}^{\infty} \mathbb{E}_{(\boldsymbol{s}_t, \boldsymbol{a}_t) \sim \rho_\pi}[r_t + \alpha \mathcal{H}_\pi[\boldsymbol{a}_t | \boldsymbol{s}_t]], \tag{11}$$

where $\mathcal{H}_\pi[\boldsymbol{a}_t|\boldsymbol{s}_t] = -\int_{\mathcal{A}} \pi(\boldsymbol{a}_t|\boldsymbol{s}_t) \log \pi(\boldsymbol{a}_t|\boldsymbol{s}_t) d\boldsymbol{a}_t$ is the continuous Shannon entropy of the action distribution, and the temperature $\alpha$ balances the entropy term versus the reward.

Incorporating the discount factor (Thomas, 2014; Haarnoja et al., 2017), we obtain the following objective which maximizes the expected return and entropy for future states starting from $(\boldsymbol{s}_t, \boldsymbol{a}_t)$ weighted by its probability $\rho_\pi$ under policy $\pi$ :

$$J_{\text{maxent}}(\pi) = \sum_{t=0}^{\infty} \mathbb{E}_{(\boldsymbol{s}_t, \boldsymbol{a}_t) \sim \rho_\pi} \left[ \sum_{l=t}^{\infty} \gamma^{l-t} \mathbb{E}_{(\boldsymbol{s}_l, \boldsymbol{a}_l) \sim \rho_\pi}[r_t + \alpha \mathcal{H}_\pi[\boldsymbol{a}_l|\boldsymbol{s}_l]] \right]. \tag{12}$$

The soft $Q$-function is the expected value under $\pi$ of the discounted sum of rewards and entropy :

$$Q_{\text{soft}}^\pi(\boldsymbol{s}_t, \boldsymbol{a}_t) = r_t + \mathbb{E}_{(\boldsymbol{s}_{t+1}, \dots) \sim \rho_\pi} \left[ \sum_{l=t+1}^{\infty} \gamma^l (r_l + \alpha \mathcal{H}_\pi[\boldsymbol{a}_l|\boldsymbol{s}_l]) \right], \tag{13}$$

and the soft value function is :

$$V_{\text{soft}}^\pi(\boldsymbol{s}_t) = \alpha \log \int_{\mathcal{A}} \exp(\frac{1}{\alpha} Q_{\text{soft}}^\pi(\boldsymbol{s}, \boldsymbol{a})) d\boldsymbol{a}. \tag{14}$$

When $\pi(\boldsymbol{a}|\boldsymbol{s}) = \exp(\frac{1}{\alpha}(Q_{\text{soft}}^\pi(\boldsymbol{s}, \boldsymbol{a}) - V_{\text{soft}}^\pi(\boldsymbol{s}))) \triangleq \pi^*$, then the soft Bellman equation yields the following relationship :

$$Q_{\text{soft}}^\pi(\boldsymbol{s}_t, \boldsymbol{a}_t) = r_t + \gamma \mathbb{E}_{(\boldsymbol{s}_{t+1}, \dots) \sim \rho_\pi}[V_{\text{soft}}^\pi(\boldsymbol{s}_{t+1})], \tag{15}$$

where we can rewrite the discounted maximum entropy objective in Eq. 12 :

$$J_{\text{maxent}}(\pi) = \sum_{t=0}^{\infty} \mathbb{E}_{(\boldsymbol{s}_t, \boldsymbol{a}_t) \sim \rho_\pi} \left[ Q_{\text{soft}}^\pi(\boldsymbol{s}, \boldsymbol{a}) + \alpha \mathcal{H}_\pi[\boldsymbol{a}_t|\boldsymbol{s}_t] \right] \tag{16}$$

$$= \sum_{t=0}^{\infty} \mathbb{E}_{(\boldsymbol{s}_t, \boldsymbol{a}_t) \sim \rho_\pi} \left[ r_t + \alpha \mathcal{H}_\pi[\boldsymbol{a}_t|\boldsymbol{s}_t] + \gamma V_{\text{soft}}^\pi(\boldsymbol{s}_{t+1}) \right]. \tag{17}$$

By Soft Policy Iteration (Haarnoja et al., 2018a), the soft Bellman operator $\mathcal{B}^*$ defined by $(\mathcal{B}^* Q)(\boldsymbol{s}_t, \boldsymbol{a}_t) = r_t + \gamma \mathbb{E}_{\boldsymbol{s}_{t+1} \sim \rho_\pi}[V(\boldsymbol{s}_{t+1})]$ has a unique contraction $Q^* = \mathcal{B}^* Q^*$ (Fox et al., 2016) and converges to the optimal policy $\pi^*$.

**Our main observation** is when the environment is a differentiable simulation, we can use FOBG estimates to directly optimize $J_{\text{maxent}}(\pi)$, including discounted policy entropy. Consider the entropy-augmented $H$-step return :

$$R_{0:H}^\alpha(\tau) = \sum_{t=0}^{H-1} \gamma^t (r_t + \alpha \mathcal{H}_\pi[\boldsymbol{a}_t|\boldsymbol{s}_t]), \tag{18}$$

then we have a single first-order estimate of Eq. 17 :

$$\hat{\nabla}_\theta^{[1]} J_{\text{maxent}}(\pi) = \nabla_\theta (R_{0:H}^\alpha + \gamma^H V_{\text{soft}}(\boldsymbol{s}_H)). \tag{19}$$

Furthermore, we can incorporate the entropy-augmented return into $TD(\lambda)$ estimates of Eq. 9 using soft value-bootstrapped $k$-step returns :

$$\Gamma_{t:t+k} = \left( \sum_{l=0}^{k-1} \gamma^l (r_{t+l} + \alpha \mathcal{H}_\pi[\boldsymbol{a}_{t+l}|\boldsymbol{s}_{t+l}]) \right) + \gamma^k V_{\text{soft}}(\boldsymbol{s}_{t+k}), \tag{20}$$

where $\tilde{V}_{\text{soft}}(\boldsymbol{s}_t) = \Gamma_{t:t+H}^\lambda$, and the value function is trained by minimizing Eq. 8 with $V_{\text{soft}}, \tilde{V}_{\text{soft}}$, and $\Gamma_{t:t+k}$ substituted in place of $V, \tilde{V}$, and $G_{t:t+k}$. We refer to this maximum entropy FO-MBRL formulation as **Soft Analytic Policy Optimization** (SAPO).

Note that we instantiate SAPO as an actor-critic algorithm that learns the soft value function by TD learning with on-policy data. In comparison, Soft Actor-Critic (SAC), a popular off-policy maximum entropy model-free RL algorithm, estimates soft $Q$-values by minimizing the soft Bellman residual with data sampled from a replay buffer. Connections may also drawn between SAPO to a maximum entropy variant of SVG($H$) (Heess et al., 2015; Amos et al., 2021), which uses rollouts from a learned world model instead of trajectories from differentiable simulation.

## 4.2 DESIGN CHOICES

**I. Entropy adjustment.** In practice, we apply automatic temperature tuning (Haarnoja et al., 2018b) to match a target entropy $\mathcal{H}$ via an additional Lagrange dual optimization step :

$$\min_{\alpha_t \geq 0} \mathbb{E}_{(\boldsymbol{s}_t, \boldsymbol{a}_t) \sim \rho_\pi}[\alpha_t(\mathcal{H}_\pi[\boldsymbol{a}_t|\boldsymbol{s}_t] - \bar{\mathcal{H}})]. \tag{21}$$

We use $\bar{\mathcal{H}} = -\dim(\mathcal{A})/2$ following (Ball et al., 2023).

**II. Target entropy normalization.** To mitigate non-stationarity in target values (Yu et al., 2022) and improve robustness across tasks with varying reward scales and action dimensions, we normalize entropy estimates. The continuous Shannon entropy is not scale invariant (Marsh, 2013). In particular, we offset (Han & Sung, 2021) and scale entropy by $\mathcal{H}$ to be approximately contained within $[0, +1]$.

**III. Stochastic policy parameterization.** We use state-*dependent* variance, with squashed Normal distribution $\pi_\theta = \tanh(\mathcal{N}(\mu_\theta(\mathbf{s}), \sigma_\theta^2(\mathbf{s})))$, which aligns with SAC (Haarnoja et al., 2018b). This enables policy entropy adjustment and captures aleatoric uncertainty in the environment (Kendall & Gal, 2017; Chua et al., 2018). In contrast, SHAC uses state-independent variance, similar to the original PPO implementation (Schulman et al., 2017).

**IV. Critic ensemble, no target networks.** We use the clipped double critic trick (Fujimoto et al., 2018) and also remove the critic target network in SHAC, similar to (Georgiev et al., 2024). However when updating the actor, we instead compute the *average* over the two value estimates to include in the return (Eq. 19), while using the *minimum* to estimate target values in standard fashion, following (Ball et al., 2023). While originally intended to mitigate overestimation bias in $Q$-learning (due to function approximation and stochastic optimization (Thrun & Schwartz, 2014)), prior work has shown that the value lower bound obtained by clipping can be overly conservative and cause the policy to pessimistically underexplore (Ciosek et al., 2019; Moskovitz et al., 2021).

Target networks (Mnih et al., 2015) are widely used (Lillicrap et al., 2016; Fujimoto et al., 2018; Haarnoja et al., 2018b) to stabilize temporal difference (TD) learning, at the cost of slower training. Efforts have been made to eliminate target networks (Kim et al., 2019; Yang et al., 2021; Shao et al., 2022; Gallici et al., 2024), and recently Cross$Q$ (Bhatt et al., 2024) has shown that careful use of normalization layers can stabilize off-policy model-free RL to enable removing target networks for improved sample efficiency. Cross$Q$ also reduces Adam $\beta_1$ momentum from 0.9 to 0.5, while keeping the default $\beta_2 = 0.999$. In comparison, SHAC uses $\beta_1 = 0.7$ and $\beta_2 = 0.95$. Using smaller momentum parameters decreases exponential decay (for the moving average estimates of the 1st and 2nd moments of the gradient) and effectively gives higher weight to more recent gradients, with less smoothing by past gradient history (Kingma & Ba, 2015).

**V. Architecture and optimization.** We use SiLU (Elfwing et al., 2018) instead of ELU for the activation function. We also switch the optimizer from Adam to AdamW (Loshchilov & Hutter, 2017), and lower gradient norm clipping from 1.0 to 0.5. Note that SHAC already uses LayerNorm (Ba et al., 2016), which has been shown to stabilize TD learning when not using target networks or replay buffers (Bhatt et al., 2024; Gallici et al., 2024).

## 5 REWARPED: PARALLEL DIFFERENTIABLE MULTIPHYSICS SIMULATION

We aim to evaluate our approach on more challenging manipulation and locomotion tasks that involve interaction between articulated rigid bodies and deformables. To this end, we introduce Rewarped, our parallel differentiable multiphysics simulation platform that provides GPU-accelerated parallel environments for RL and enables computing batched simulation gradients efficiently. We build Rewarped on NVIDIA Warp (Macklin, 2022), the successor to DFlex (Xu et al., 2021; Murthy et al., 2021; Turpin et al., 2022; Heiden et al., 2023).

We proceed to discuss high-level implementation details and optimization tricks to enable efficient parallel differentiable simulation. We develop a parallelized implementation of Material Point Method (MPM) which supports simulating parallel environments of complex deformable materials, building on the MLS-MPM implementation by (Ma et al., 2023) used for non-parallel simulation. Furthermore, we support one-way coupling from kinematic articulated rigid bodies to MPM particles, based on the (non-parallel) MPM-based simulation from (Huang et al., 2020; Li et al., 2023a).

## 5.1 PARALLEL DIFFERENTIABLE SIMULATION

We implement all simulation code in NVIDIA Warp (Macklin, 2022), a library for differentiable programming that converts Python code into CUDA kernels by runtime JIT compilation. Warp implements reverse-mode auto-differentiation through the discrete adjoint method, using a tape to record kernel calls for the computation graph, and generates kernel adjoints to compute the backward pass. Warp uses source-code transformation (Griewank & Walther, 2008; Hu et al., 2020) to automatically generate kernel adjoints.

We use gradient checkpointing (Griewank & Walther, 2000; Qiao et al., 2021b) to reduce memory requirements. During backpropagation, we run the simulation forward pass again to recompute intermediate values, instead of saving them during the initial forward pass. This is implemented by capturing and replaying CUDA graphs, for both the forward pass and the backward pass of the simulator. Gradient checkpointing by CUDA graphs enables us to compute batched simulation gradients over multiple time steps efficiently, when using more simulation substeps for simulation stability. We use a custom PyTorch autograd function to interface simulation data and model parameters between Warp and PyTorch while maintaining auto-differentiation functionality.

## 6 EXPERIMENTS

We evaluate our proposed maximum entropy FO-MBRL algorithm, Soft Analytic Policy Optimization (SAPO, Section 4), against baselines on a range of locomotion and manipulation tasks involving rigid and soft bodies. We implement these tasks in Rewarped (Section 5), our parallel differentiable multiphysics simulation platform. We also compare algorithms on DFlex rigid-body locomotion tasks introduced in (Xu et al., 2021) in Appendix F.2.

**Baselines.** We compare to vanilla model-free RL algorithms: Proximal Policy Optimization (PPO, Schulman et al. (2017)), an on-policy actor-critic algorithm; Soft Actor-Critic (SAC, Haarnoja et al. (2018b)) an off-policy maximum entropy actor-critic algorithm. We use the implementations and hyperparameters from (Li et al., 2023b) for both, which have been validated to scale well with parallel simulation. Implementation details (network architecture, common hyperparameters, etc.) are standardized between methods for fair comparison, see Appendix C. We also compare against Analytic Policy Gradient (APG, Freeman et al. (2021)) and Short-Horizon Actor-Critic (SHAC, Xu et al. (2021)), both of which are state-of-the-art FO-MBRL algorithms that leverage first-order analytic gradients from differentiable simulation for policy learning. Finally, we include gradient-based trajectory optimization (TrajOpt) as a baseline, which uses differentiable simulation gradients to optimize for an open-loop action sequence that maximizes total rewards across environments.

**Tasks**. Using Rewarped, we re-implement a range of challenging manipulation and locomotion tasks involving rigid and soft bodies that have appeared in prior works. Rewarped enables training algorithms on parallel environments, and differentiable simulation to compute analytic simulation gradients through environment dynamics and rewards. We visualize these tasks in Figure 1. To simulate deformables, we use $\sim 2500$ particles per env. See Appendix E for more details.

**AntRun** – Ant locomotion task from DFlex (Xu et al., 2021), where the objective is to maximize the forward velocity of a four-legged ant rigid-body articulation.

**HandReorient** – Allegro hand manipulation task from Isaac Gym (Makoviychuk et al., 2021), where the objective is to perform in-hand dexterous manipulation to rotate a rigid cube given a target pose. We replace non-differentiable terms of the reward function (ie. boolean comparisons) with differentiable alternatives to enable computing analytic gradients.

**RollingFlat** – Rolling pin manipulation task from PlasticineLab (Huang et al., 2020), where the objective is to flatten a rectangular piece of dough using a cylindrical rolling pin.

**SoftJumper** – Soft jumping locomotion task, inspired by GradSim (Murthy et al., 2021) and DiffTaichi (Hu et al., 2020), where the objective is to maximize the forward velocity and height of a high-dimensional actuated soft elastic quadruped.

**HandFlip** – Shadow hand flip task from DexDeform (Li et al., 2023a), where the objective is to flip a cylindrical piece of dough in half within the palm of a dexterous robot hand.

**FluidMove** – Fluid transport task from SoftGym (Lin et al., 2021), where the objective is to move a container filled with fluid to a given target position, without spilling fluid out of the container.

Note that {AntRun, HandReorient} are tasks that involve articulated rigid bodies only, with state-based observations. In contrast, {RollingFlat, SoftJumper, HandFlip, FluidMove} are tasks that also involve deformables, with both state-based and high-dimensional (particle-based) visual observations.

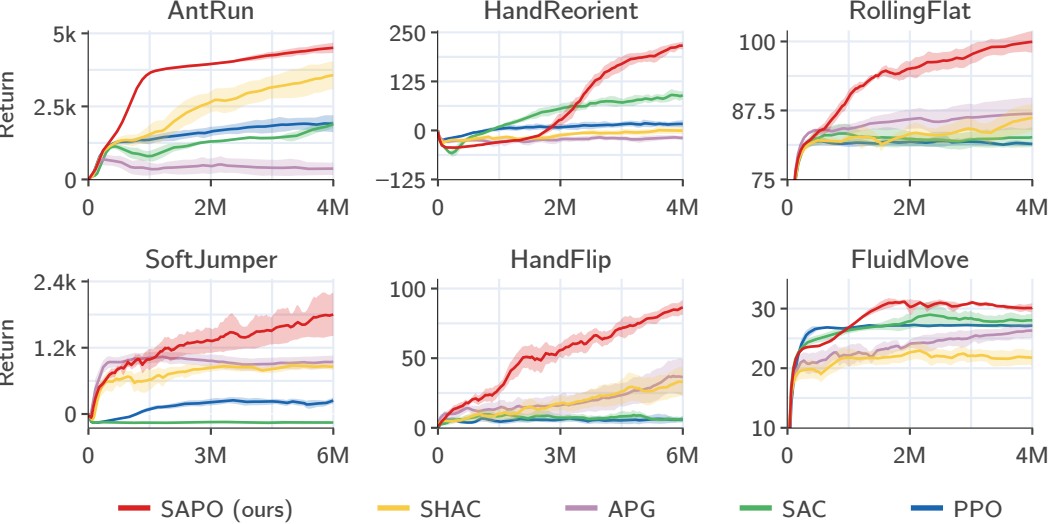

*Figure 2:* **Rewarped tasks training curves.** Episode return as a function of environment steps in Rewarped AntRun ($\mathcal{A} \subset \mathbb{R}^8$), HandReorient ($\mathcal{A} \subset \mathbb{R}^{16}$), RollingFlat ($\mathcal{A} \subset \mathbb{R}^3$), SoftJumper ($\mathcal{A} \subset \mathbb{R}^{222}$), HandFlip ($\mathcal{A} \subset \mathbb{R}^{24}$), and FluidMove ($\mathcal{A} \subset \mathbb{R}^3$) tasks. Smoothed using EWMA with $\alpha = 0.99$. Mean and 95% CIs over 10 random seeds.

|  | **AntRun** | **HandReorient** | **RollingFlat** | **SoftJumper** | **HandFlip** | **FluidMove** |
|---|---|---|---|---|---|---|
| PPO | $2048.7 \pm 36.6$ | $5.9 \pm 4.9$ | $81.2 \pm 0.1$ | $261.5 \pm 12.4$ | $7.3 \pm 1.1$ | $27.3 \pm 0.2$ |
| SAC | $2063.6 \pm 13.9$ | $70.5 \pm 10.2$ | $83.0 \pm 0.3$ | $-161.8 \pm 2.5$ | $4.6 \pm 1.1$ | $28.2 \pm 0.7$ |
| TrajOpt | $915.5 \pm 29.6$ | $-12.5 \pm 2.0$ | $81.5 \pm 0.1$ | $437.2 \pm 17.7$ | $27.3 \pm 2.6$ | $27.0 \pm 0.1$ |
| APG | $258.7 \pm 20.3$ | $-11.6 \pm 1.9$ | $86.9 \pm 0.4$ | $956.6 \pm 15.6$ | $38.2 \pm 3.5$ | $26.3 \pm 0.3$ |
| SHAC | $3621.0 \pm 54.4$ | $-2.5 \pm 1.8$ | $86.8 \pm 0.4$ | $853.3 \pm 10.2$ | $32.7 \pm 2.9$ | $21.7 \pm 0.4$ |
| SAPO (ours) | $4535.9 \pm 24.5$ | $221.7 \pm 9.5$ | $100.4 \pm 0.4$ | $1820.5 \pm 47.9$ | $90.0 \pm 2.2$ | $30.6 \pm 0.4$ |

*Table 2:* **Rewarped tasks tabular results.** Evaluation episode returns for final policies after training. Mean and 95% CIs over 10 random seeds with $2N$ episodes per seed for $N = 32$ or $64$ parallel envs.

## 6.1 RESULTS ON REWARPED TASKS

We compare SAPO, our proposed maximum entropy FO-MBRL algorithm, against baselines on a range of challenging manipulation and locomotion tasks that involve rigid and soft bodies, re-implemented in Rewarped, our parallel differentiable multiphysics simulation platform. In Figure 2, we visualize training curves to compare algorithms. SAPO shows better training stability across different random seeds, against existing FO-MBRL algorithms APG and SHAC. In Table 2, we report evaluation performance for final policies after training. SAPO outperforms all baselines across all tasks we evaluated, given the same budget of total number of environment steps. We also find that on tasks involving deformables, APG outperforms SHAC, which is consistent with results in DaXBench (Chen et al., 2022) on their set of soft-body manipulation tasks. However, SHAC outperforms APG on the articulated rigid-body tasks, which agrees with the rigid-body locomotion results in DFlex (Xu et al., 2021) that we also reproduce ourselves in Appendix F.2.

In Appendix Figure 11, we visualize different trajectories produced by SAPO policies after training. We observe that SAPO learns to perform tasks with deformables that we evaluate on. For RollingFlat, SAPO controls the rolling pin to flatten the dough and spread it across the ground. For SoftJumper, SAPO learns a locomotion policy that controls a soft elastic quadruped to jump forwards. For HandFlip, SAPO is capable of controlling a high degree-of-freedom dexterous robot hand, to flip a

piece of dough in half within the palm of the hand. For FluidMove, SAPO learns a policy to move the container of fluid with minimal spilling. Additionally, SAPO learns a successful locomotion policy for the articulated rigid-body locomotion task AntRun. For HandReorient however, SAPO is only capable of catching the cube and preventing it from falling to the ground. This is a challenging task that will likely require more environment steps to learn policies capable of re-orienting the cube to given target poses in succession.

## 6.2 SAPO ABLATIONS

We investigate which components of SAPO yield performance gains over SHAC, on the HandFlip task. We conduct several ablations on SAPO: (a) w/o $V_{\text{soft}}$, where instead the critic is trained in standard fashion without entropy in target values; (b) w/o $\mathcal{H}_\pi$, where we do not use entropy-augmented returns and instead train the actor to maximize expected returns only; (c) w/o $\mathcal{H}_\pi$ and $V_{\text{soft}}$, which corresponds to modifying SHAC by applying design choices {III, IV, V} described in Section 4.2.

We visualize training curves in Figure 3, and in Table 3 we report final evaluation performance as well as percentage change from SHAC's performance as the baseline. From ablation (b), we find that using analytic gradients to train a policy to maximize both expected return and entropy is critical to the performance of SAPO, compared to ablation (a) which only replaces the soft value function.

Additionally, we observe that ablation (c), where we apply design choices {III, IV, V} onto SHAC, result in approximately half of the performance improvement of SAPO over SHAC on the HandFlip task. We also conducted this ablation on the DFlex rigid-body locomotion tasks however, and found these modifications to SHAC to have minimal impact in those settings, shown in Appendix F.3. We also conduct individual ablations for these three design choices in Appendix F.4.

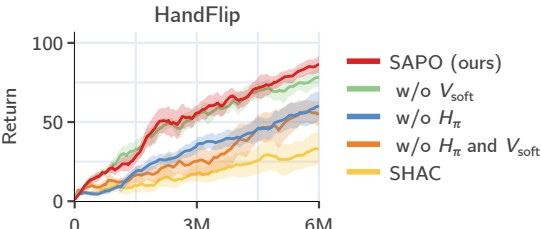

| | **HandFlip** | $(\Delta\%)$ |
|---|---|---|
| SAPO (ours) | $90 \pm 2$ | +172.7% |
| w/o $V_{\text{soft}}$ | $77 \pm 3$ | +133.3% |
| w/o $\mathcal{H}_\pi$ | $59 \pm 4$ | +78.8% |
| w/o $\mathcal{H}_\pi$ and $V_{\text{soft}}$ | $56 \pm 3$ | +69.7% |
| SHAC | $33 \pm 3$ | – |

*Figure 3:* **SAPO ablations – HandFlip training curves.** Episode return as a function of environment steps. Smoothed using EWMA with $\alpha = 0.99$. Mean and 95% CIs over 10 random seeds.

*Table 3:* **SAPO ablations – HandFlip tabular results.** Evaluation episode returns for final policies after training. Mean and 95% CIs over 10 random seeds with 64 episodes per seed.

## 7 CONCLUSION

Due to high sample complexity requirements and slower runtimes for soft-body simulation, RL has had limited success on tasks involving deformables. To address this, we introduce Soft Analytic Policy Optimization (SAPO), a first-order model-based actor-critic RL algorithm based on the maximum entropy RL framework, which leverages first-order analytic gradients from differentiable simulation to achieve higher sample efficiency. Alongside this approach, we present Rewarped, a scalable and easy-to-use platform which enables parallelizing RL environments of GPU-based differentiable multiphysics simulation. We re-implement challenging locomotion and manipulation tasks involving rigid bodies, articulations, and deformables using Rewarped. On these tasks, we demonstrate that SAPO outperforms baselines in terms of sample efficiency as well as task performance given the same budget for total environment steps.

**Limitations.** SAPO relies on end-to-end learning using first-order analytic gradients from differentiable simulation. Currently, we use (non-occluded) subsampled particle states from simulation as observations to policies, which is infeasible to obtain in real-world settings. Future work may use differentiable rendering to provide more realistic visual observations for policies while maintaining differentiability, towards sim2real transfer of policies learned using SAPO. Another promising direction to consider is applications between differentiable simulation and learned world models.

ACKNOWLEDGMENTS

The authors would like to thank Uksang Yoo, Ananye Agarwal, and David Held for helpful discussions on this work. We would also like to thank Miles Macklin, Eric Heiden, and the rest of the NVIDIA Warp team. This work was in part supported by the Technology Innovation Program (20018112, Development of autonomous manipulation and gripping technology using imitation learning based on visual-tactile sensing) funded by the Ministry of Trade, Industry & Energy (MOTIE), Korea.

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

## A    EXTENDED RELATED WORK

**Differentiable simulators.** We review prior works on *non-parallel* differentiable simulators for robotics and control problems. Degrave et al. (2019) demonstrate that gradient-based optimization can be used to train neural network controllers by implementing a rigid-body physics engine in a modern auto-differentiation library to compute analytic gradients. de Avila Belbute-Peres et al. (2018) show how to analytically differentiate through 2D rigid-body dynamics, by defining a linear complementarity problem (LCP) and using implicit differentiation. Exploiting sparsity to differentiate through the LCP more efficiently, Nimble (Werling et al., 2021) presents a differentiable rigid-body physics engine that supports complex 3D contact geometries. Dojo (Howell et al., 2022) implements a differentiable rigid-body physics engine with a hard contact model (based on nonlinear friction cones) via a nonlinear complementarity problem (NCP), and a custom interior-point solver to reliably solve the NCP. Also using implicit differentiation, Qiao et al. (2020) adopt meshes as object representations to support both rigid and deformable objects and adopt a sparse local collision handling scheme that enables coupling between objects of different materials. Geilinger et al. (2020) presents a unified frictional contact model for rigid and deformable objects, while analytically computing gradients through adjoint sensitivity analysis instead. DiffTaichi (Hu et al., 2019a; 2020), used within several differentiable soft-body simulators (Hu et al., 2019b; Huang et al., 2020; Zhou et al., 2023; Si et al., 2024; Wang et al., 2024), introduces a differentiable programming framework for tape-based auto-differentiation using source-code transformation to generate kernel adjoints. We build Rewarped using NVIDIA Warp (Macklin, 2022), a differentiable programming framework similar to DiffTaichi.

**Model-based reinforcement learning.** Generally, approaches in MBRL (Moerland et al., 2023) either assume a known (non-differentiable) model or learn a world model by predicting dynamics and rewards from data. We will focus on the latter, as learned models can be differentiated through for policy learning, similar to FO-MBRL which computes analytic policy gradients by backpropagating rewards from some known differentiable model (ie. differentiable simulator). Value gradient methods estimate policy gradients by directly backpropagating through the learned model (Deisenroth et al., 2013; Heess et al., 2015; Feinberg et al., 2018; Parmas et al., 2018; Clavera et al., 2020; Hafner et al., 2020). Learned models can also be used for data augmentation by generating imagined trajectories from the learned model, as first proposed in DYNA (Sutton, 1990). In this paper, we consider FO-MBRL (Freeman et al., 2021; Xu et al., 2021) which does analytic policy optimization with data and gradients from a differentiable simulator, without explicitly learning a world model.

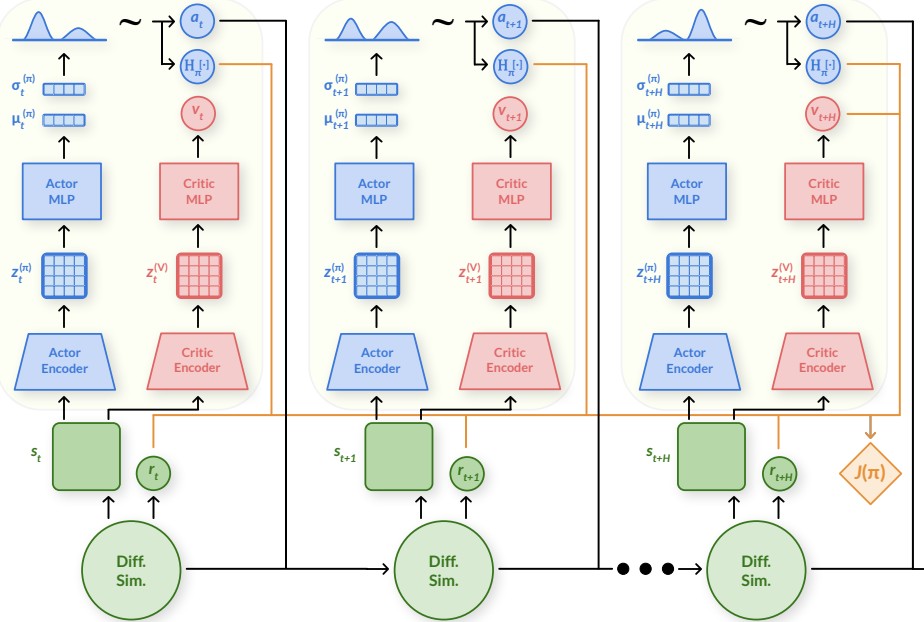

*Figure 4:* **Diagram of computational graph of SAPO.** Note that $J(\pi)$ only updates the actor, although gradients are still propagated through the critic.

# B  SAPO ALGORITHM DETAILS

## B.1  VISUAL ENCODERS IN DIFFERENTIABLE SIMULATION

We use separate visual encoders for the actor $\pi_\theta(\boldsymbol{a}_t|f_\phi(\boldsymbol{s}_t))$ and critic $V_\psi(f_\zeta(\boldsymbol{s}_t))$, to enable learning on deformable tasks with high-dimensional (particle-based) visual inputs. To maintain differentiability to compute analytic gradients and reduced memory requirements, we use downsampled particle states of the simulation as point clouds for high-dimensional visual observations. For runtime efficiency, we use a PointNet-like DP3 encoder (Ze et al., 2024) to encode a point cloud observation into a lower-dimensional latent vector. We leave combining differentiable rendering (of RGB or depth image observations) with differentiable simulation, like in (Murthy et al., 2021), to future work.

## B.2  PSEUDOCODE

---

**Algorithm 1:** Soft Analytic Policy Optimization (SAPO)

---

Initialize network parameters $\theta, \phi, \psi_i, \zeta_i$
$t_0 \leftarrow 0$
**repeat**
    Create buffer $\mathcal{B}$
    **for** $t = t_0 + 1 \dots H$ **do**
        $\boldsymbol{a}_t \sim \pi_\theta(\cdot|f_\phi(\boldsymbol{s}_t))$
        $h_t \leftarrow \mathcal{H}_\pi[\boldsymbol{a}_t|\boldsymbol{s}_t]$
        $\hat{h}_t \leftarrow (h_t + |\bar{\mathcal{H}}|)/(2|\bar{\mathcal{H}}|)$
        $\boldsymbol{s}_{t+1}, r_t, d_t \leftarrow \texttt{env.step}(\boldsymbol{a}_t)$
        $v_{t+1}^{(i)} \leftarrow V_{\psi_i}(f_{\zeta_i}(\boldsymbol{s}_{t+1}))$
        **if** $d_t$ **then**
            $t_0 \leftarrow 0$
        ▷ Add data to buffer :
        $\mathcal{B} \leftarrow \mathcal{B} \cup \{(\boldsymbol{s}_t, \boldsymbol{a}_t, r_t, d_t, h_t, \{v_{t+1}^{(i)}\})\}$
    $t_0 \leftarrow t_0 + (H + 1)$

    ▷ Update actor using Eq. 17, with normalized
     entropy $\hat{h}_t$ and mean values $\frac{1}{C}\sum_{i=1}^{C} v_t^{(i)}$ :
    $(\theta, \phi) \leftarrow (\theta, \phi) - \eta \nabla_{(\theta,\phi)} J_{\text{maxent}}(\pi)$

    ▷ Detach data from differentiable simulation
     autograd :
    $\mathcal{B} \leftarrow \texttt{stopgrad}(\mathcal{B})$

    ▷ Update entropy temperature using Eq. 21,
     with unnormalized entropy $h_t$ :
    $\alpha \leftarrow \alpha - \eta \nabla_\alpha[\frac{1}{H}\sum_{t=1}^{H} \alpha(h_t - \bar{\mathcal{H}})]$

    ▷ Compute TD($\lambda$) value targets via Eq. 9
     using soft returns of Eq. 20, with
     normalized entropy $\hat{h}_t$ and min values
     $\min_{i=1\dots C} v_t^{(i)}$ :
    $\tilde{v}_t \leftarrow \dots$

    **for** $K$ *updates* **do**
        Sample $(\boldsymbol{s}_t, \tilde{v}_t) \sim \mathcal{B}$
        ▷ Update critics using Eq. 8 with clipped
         soft value targets $\tilde{v}$ :
        $(\psi_i, \zeta_i) \leftarrow (\psi_i, \zeta_i) - \eta \nabla_{(\psi_i,\zeta_i)} \mathcal{L}(V)$
**until** *converged*;

**Model components**

| | |
|---|---|
| Actor | $\pi_\theta(\boldsymbol{a}_t|f_\phi(\boldsymbol{s}_t))$ |
| Actor encoder | $f_\phi(\boldsymbol{s}_t)$ |
| Critic | $V_{\psi_i}(f_{\zeta_i}(\boldsymbol{s}_t))$ |
| Critic encoder | $f_{\zeta_i}(\boldsymbol{s}_t)$ |
| Critic index | $i = 1 \dots C$ |

**Hyperparameters**

| | |
|---|---|
| Horizon | $H$ |
| Entropy temperature | $\alpha$ |
| Target entropy | $\bar{\mathcal{H}}$ |
| TD trace decay | $\lambda$ |
| Discount | $\gamma$ |
| Learning rates | $\eta$ |
| Num critics | $C$ |
| Mini-epochs | $K$ |

---

## C  HYPERPARAMETERS

We run all algorithms on consumer workstations with NVIDIA RTX 4090 GPUs. Each run uses a single GPU, on which we run both the GPU-accelerated parallel simulation and optimization loop. We use a recent high-performance implementation of standard model-free RL algorithms which has been validated for parallel simulation (Li et al., 2023b). We aim to use common hyperparameter values among algorithms where applicable, such as for discount factor, network architecture, etc.

For TrajOpt, we initialize a single $T$-length trajectory of zero actions. This action is repeated across $N = 16$ parallel environments ($N = 32$ for {AntRun, HandReorient}). We optimize this trajectory for 50 epochs with a horizon $H$ of 32 steps. We use AdamW as the optimizer, with learning rate of 0.01, decay rates $(\beta_1, \beta_2) = (0.7, 0.95)$, and gradient norm clipping of 0.5. For evaluation, we playback this single trajectory across parallel environments, each with different random initial states.

| | *shared* | PPO | SAC | APG | SHAC | SAPO |
|---|---|---|---|---|---|---|
| Num envs $N$ | 64 | | | | | |
| Batch size | 2048 | | | | | |
| Horizon $H$ | 32 | | | | | |
| Mini-epochs $K$ | | 5 | 8 | 1 | 16 | 16 |
| Discount $\gamma$ | 0.99 | | | | | |
| TD/GAE $\lambda$ | - | 0.95 | | | 0.95 | 0.95 |
| Actor $\eta$ | | 5e−4 | 5e−4 | 2e−3 | 2e−3 | 2e−3 |
| Critic $\eta$ | - | 5e−4 | 5e−4 | | 5e−4 | 5e−4 |
| Entropy $\eta$ | - | | 5e−3 | | | 5e−3 |
| $\eta$ schedule | - | KL(0.008) | | linear | linear | linear |
| Optim type | AdamW | | | Adam | Adam | |
| Optim $(\beta_1, \beta_2)$ | (0.9, 0.999) | | | (0.7, 0.95) | (0.7, 0.95) | (0.7, 0.95) |
| Grad clip | 0.5 | | | 1.0 | 1.0 | |
| Norm type | LayerNorm | | | | | |
| Act type | SiLU | | | ELU | ELU | |
| Actor $\sigma(\mathbf{s})$ | yes | | | no | no | |
| Actor $\log(\sigma)$ | - | $\log[0.1, 1.0]$ | $[-5, 2]$ | | | $[-5, 2]$ |
| Num critics $C$ | - | | 2 | | | 2 |
| Critic $\tau$ | - | | 0.995 | | 0.995 | |
| Replay buffer | - | | $10^6$ | | | |
| Target entropy $\bar{\mathcal{H}}$ | - | | $-\dim(\mathcal{A})/2$ | | | $-\dim(\mathcal{A})/2$ |
| Init temperature | - | | 1.0 | | | 1.0 |

*Table 4:* **Shared hyperparameters.** Algorithms use hyperparameter settings in the *shared* column unless otherwise specified in an individual column.

| | Hopper | Ant | Humanoid | SNUHumanoid |
|---|---|---|---|---|
| Actor MLP | $(128, 64, 32)$ | $(128, 64, 32)$ | $(256, 128)$ | $(512, 256)$ |
| Critic MLP | $(64, 64)$ | $(64, 64)$ | $(128, 128)$ | $(256, 256)$ |

*Table 5:* **DFlex task-specific hyperparararameters.** All algorithms use the same actor and critic network architecture.

| | *shared* | AntRun | HandReorient |
|---|---|---|---|
| Num envs $N$ | 32 | 64 | 64 |
| Batch size | 1024 | 2048 | 2048 |
| Actor MLP | $(512, 256)$ | $(128, 64, 32)$ | |
| Critic MLP | $(256, 128)$ | $(64, 64)$ | |

*Table 6:* **Rewarped task-specific hyperparararameters.** All algorithms use the same actor and critic network architecture. Algorithms use hyperparameter settings in the *shared* column unless otherwise specified in an individual column.

# D    REWARPED PHYSICS

We review the simulation techniques used to simulate various rigid and soft bodies in Rewarped. Our discussion is based on (Xu et al., 2021; Heiden et al., 2023; Murthy et al., 2021; Macklin, 2022), as well as (Hu et al., 2019b; Huang et al., 2020; Ma et al., 2023) for MPM.

To backpropagate analytic policy gradients, we need to compute simulation gradients. Following reverse order, observe :

$$\nabla_\theta J(\pi) = \nabla_\theta \sum_{t=0}^{T-1} \gamma^t r_t \tag{22}$$

$$\frac{\partial r_t}{\partial \theta} = \frac{\partial r_t}{\partial \boldsymbol{a}_t} \frac{\partial \boldsymbol{a}_t}{\partial \theta} + \frac{\partial r_t}{\partial \boldsymbol{s}_t} \frac{d\boldsymbol{s}_t}{d\theta} \tag{23}$$

$$\frac{d\boldsymbol{s}_{t+1}}{d\theta} = \frac{\partial f(\boldsymbol{s}_t, \boldsymbol{a}_t)}{\partial \boldsymbol{s}_t} \frac{d\boldsymbol{s}_t}{d\theta} + \frac{\partial f(\boldsymbol{s}_t, \boldsymbol{a}_t)}{\partial \boldsymbol{s}_t} \frac{\partial \boldsymbol{a}_t}{\partial \theta} \tag{24}$$

$$\text{with } \boldsymbol{a}_t \sim \pi_\theta(\cdot | \boldsymbol{s}_t), \boldsymbol{s}_{t+1} = f(\boldsymbol{s}_t, \boldsymbol{a}_t). \tag{25}$$

We use the reparameterization trick to compute gradients $\frac{\partial \pi_\theta(\boldsymbol{s})}{\partial \theta}$ for stochastic policy $\pi_\theta$ to obtain $\frac{\partial \boldsymbol{a}_t}{\partial \theta}$. Note that the simulation gradient is $\frac{\partial f(\boldsymbol{s}_t, \boldsymbol{a}_t)}{\partial \boldsymbol{s}_t}$, which we compute by the adjoint method through auto-differentiation (AD).

While we could estimate the simulation gradient with finite differences (FD) through forms of $f'(x) \approx \frac{f(x+h)-f(x)}{h}$, FD has two major limitations. Consider $f : \mathbb{R}^N \to \mathbb{R}^M$, where $N$ is the input dimension, $M$ is the output dimension, and $L$ is the cost of evaluating $f$. Then FD has time complexity $O((N+1) \cdot M \cdot L)$. In comparison, reverse-mode AD has time complexity $O(M \cdot L)$, and also benefits from GPU-acceleration. Furthermore, FD only computes numerical approximations of the gradient with precision depending on the value of $h$, while AD yields analytic gradients. For a simulation platform built on a general differentiable programming framework, computing gradients for *different* simulation dynamics $f$ is straightforward using AD. While alternatives to compute analytic simulation gradients through implicit differentiation have been proposed (see Appendix A), they are less amenable to batched gradient computation for parallel simulation.

## D.1    COMPOSITE RIGID BODY ALGORITHM (CRBA)

Consider the following rigid body forward dynamics equation to solve :

$$M\ddot{q} = J^T \mathcal{F}(q, \dot{q}) + c(q, \dot{q}) + \tau(q, \dot{q}, a), \tag{26}$$

where $q, \dot{q}, \ddot{q}$ are joint coordinates, velocities, accelerations, $\mathcal{F}$ are external forces, $c$ includes Coriolis forces, $\tau$ are joint-space actuations, $M$ is the mass matrix, and $J$ is the Jacobian. Featherstone's composite rigid body algorithm (CRBA) is employed to solve for articulation dynamics. After obtaining joint accelerations $\ddot{q}$, a semi-implicit Euler integration step is performed to update the system state $\boldsymbol{s} = (q, \dot{q})$. We use the same softened contact model as (Xu et al., 2021).

## D.2    FINITE ELEMENT METHOD (FEM)

To simulate dynamics, a finite element model (FEM) is employed based on tetrahedral discretization of the solid's mesh. A stable neo-Hookean constitutive model (Smith et al., 2018) is used to model elastic deformable solids with per-element actuation :

$$\Psi(q, \tau) = \frac{\mu}{2}(I_C - 3) + \frac{\lambda}{2}(J - \alpha)^2 - \frac{\mu}{2}\log(I_C + 1), \tag{27}$$

where $(\lambda, \mu)$ are the Lamé parameters which control each tetrehedral element's resistance to shear and strain, $\alpha$ is a constant, $J = \det(\mathcal{F})$ is the relative volume change, $I_C = \operatorname{tr}(\mathcal{F}^T \mathcal{F})$, and $\mathcal{F}$ is the deformation gradient. Integrating $\Psi$ over each tetrahedral element yields the total elastic potential energy, to then compute $\mathcal{F}_{elastic}$ from the energy gradient, and finally update the system state using a semi-implicit Euler integration step. We use the same approach as (Murthy et al., 2021).

### D.3 MATERIAL POINT METHOD (MPM)

The moving least squares material point method (MLS-MPM) (Hu et al., 2018) can efficiently simulate a variety of complex deformables, such as elastoplastics and liquids. Consider the two equations for conservation of momentum and mass :

$$\rho\ddot{\phi} = \nabla \cdot P + \rho b \tag{28}$$

$$\frac{D\rho}{Dt} = -\rho\nabla \cdot \dot{\phi}, \tag{29}$$

where $\dot{\phi}$ is the velocity, $\ddot{\phi}$ is the acceleration, $\rho$ is the density, and $b$ is the body force. For this system to be well-defined, constitutive laws must define $P$, see (Ma et al., 2023). We simulate elastoplastics using an elastic constitutive law with von Mises yield criterion :

$$P(F) = U(2\mu\epsilon + \lambda\mathrm{tr}(\epsilon))U^T \tag{30}$$

$$\delta\gamma = \|\hat{\epsilon}\| - \frac{2\sigma_y}{2\mu} \tag{31}$$

$$\mathcal{P}(F) = \begin{cases} F & \delta\gamma \leq 0 \\ U\exp(\epsilon - \delta\gamma\frac{\hat{\epsilon}}{\|\hat{\epsilon}\|})V^T & \delta\gamma > 0, \end{cases} \tag{32}$$

where $\mathcal{P}(F)$ is a projection back into the elastic region for stress that violates the yield criterion, and $F = U\Sigma V^T$ is the singular value decomposition (SVD). We simulate fluids as weakly compressible, using a plastic constitutive law with the fixed corotated elastic model :

$$P(F) = \lambda J(J-1)F^{-T} \tag{33}$$

$$\mathcal{P}(F) = J^{\frac{1}{3}}I, \tag{34}$$

where $\mathcal{P}(F)$ is a projection back into the plastic region and $J = \det(F)$.

We use the same softened contact model from PlasticineLab (Huang et al., 2020), with one-way coupling between rigid bodies to MPM particles. For any grid point with signed distance $d$ to the nearest rigid body, we compute a smooth collision strength $s = \min(\exp(-\alpha d), 1)$. The grid point velocity before and after collision projection is linearly interpolated using $s$. Coupling is implemented by the Compatible Particle-In-Cell (CPIC) algorithm, see steps 2–4 of (Hu et al., 2018).

# E    REWARPED TASKS

We visualize each task in Figure 1, and high-level task descriptions are provided in Section 6.

## E.1    TASK DEFINITIONS

For each task, we define the observation space $\mathcal{O}$, action space $\mathcal{A}$, reward function $R$, termination $d$, episode length $T$, initial state distribution $\rho_0$, and simulation method used for transition function $P$. We denote the goal state distribution $\rho^*$ if the task defines it. We also report important physics hyperparameters, including both shared ones and those specific to the simulation method used.

We use $\hat{i}, \hat{j}, \hat{k}$ for standard unit vectors and $\text{proj}_w u = \frac{u \cdot w}{\|w\|^2} w$ for the projection of $u$ onto $w$. We use $u_x$ to denote the $x$-coordinate of vector $u$.

We use $\mathbb{1}$ for the indicator function and $U(b) = U(-b, b)$ for the uniform distribution. We denote an additive offset $^+\delta$ or multiplicative scaling $^\times\delta$, from some prior value.

For notation, we use $z$-axis up. Let $q, \dot{q}, \ddot{q}$ be joint coordinates, velocities, accelerations. Let $(p, \theta)$ be world positions and orientations, $(v, \omega)$ be linear and angular velocities, $(a, \alpha)$ be linear and angular accelerations, derived for root links such as the center of mass (CoM).

All physical quantities are reported in SI units. We use $h$ for the frame time, and each frame is computed using $S$ substeps, so the physics time step is $\Delta t = \frac{h}{S}$. The gravitational acceleration is $g$.

**FEM.** We use $\boldsymbol{x}'$ to denote a subsampled set of FEM particle positions $\boldsymbol{x}$, where $\overline{\boldsymbol{x}'}$ is the CoM of the subsampled particles (i.e. average particle position for uniform density). Similarly for velocities $\boldsymbol{v}$. We also report the following quantities: number of particles $N_{\boldsymbol{x}}$, number of tetrahedron $N_{tet}$, particles' density $\rho$, Lamé parameters $(\lambda, \mu)$ and damping stiffness $k_{damp}$.

**MPM.** We use $\boldsymbol{x}'$ to denote a subsampled set of MPM particle positions $\boldsymbol{x}$, where $\overline{\boldsymbol{x}'}$ is the CoM of the subsampled particles (i.e. average particle position for uniform density). We also report the following quantities: friction coefficient for rigid bodies $\mu_b$, grid size $N_g$, number of particles $N_{\boldsymbol{x}}$, particles' density $\rho$, Young's modulus $E$, Poisson's ratio $\nu$, yield stress $\sigma_y$.

### E.1.1 ANTRUN

| | |
|---|---|
| $\mathcal{O}$ | $\mathbb{R}^{37} : [p_z, \theta, v, \omega, q, \dot{q}, u_{up}, u_{heading}, \boldsymbol{a}_{t-1}]$ |
| $\mathcal{A}$ | $\mathbb{R}^8$ : absolute joint torques $\tau$ |
| $R$ | $v_x + (0.1R_{up} + R_{heading} + R_{height})$ |
| $d$ | $\mathbb{1}\{p_z < h_{term}\}$ |
| $T$ | 1000 |
| $\rho_0$ | $^{+}\delta p \sim U(0.1),\, ^{+}\delta\theta \sim U(\frac{\pi}{24})$ 
 $^{+}\delta q \sim U(0.2)$ 
 $\dot{q} \sim U(0.25)$ |
| $P$ | CRBA |
| $h$ | 1/60 |
| $S$ | 16 |

*Table 7:* **AntRun task definition**. Based on Ant from (Xu et al., 2021).

The derived quantities correspond to the ant's CoM. We use termination height $h_{term} = 0.27$. We denote $u_{up} = \text{proj}_{\hat{k}}\, p$ and $u_{heading} = \text{proj}_{\hat{i}}\, p$. The reward function maximizes the forward velocity $v_x$, with auxiliary reward terms $R_{up} = u_{up}$ to encourage vertical stability, $R_{heading} = u_{heading}$ to encourage running straight, and $R_{height} = p_z - h_{term}$ to discourage falling. Initial joint angles, joint velocities, and CoM transform of the ant are randomized.

### E.1.2  HANDREORIENT

| | |
|---|---|
| $\mathcal{O}$ | $\mathbb{R}^{72}$ : "full", see (Makoviychuk et al., 2021) |
| $\mathcal{A}$ | $\mathbb{R}^{16}$ : absolute joint torques $\tau$ |
| $R$ | see (Makoviychuk et al., 2021) |
| $d$ | see (Makoviychuk et al., 2021) |
| $T$ | 600 |
| $\rho_0$ | see (Makoviychuk et al., 2021) |
| $P$ | CRBA |
| $h$ | 1/120 |
| $S$ | 32 |

*Table 8:* **HandReorient task definition**. Based on AllegroHand from (Makoviychuk et al., 2021).

See (Makoviychuk et al., 2021). Initial joint angles of the hand, CoM transform of the cube, and target orientations for the cube are randomized.

### E.1.3 ROLLINGFLAT

| | |
|---|---|
| $\mathcal{O}$ | $[\mathbb{R}^{250 \times 3}, \mathbb{R}^3, \mathbb{R}^3] : [\boldsymbol{x}', \overline{\boldsymbol{x}'}, (p_x, p_z, \theta_z)]$ |
| $\mathcal{A}$ | $\mathbb{R}^3$ : relative positions $p_x, p_z$ and orientation $\theta_z$ |
| $R$ | $R_d + R_{flat}$ |
| $d$ | - |
| $T$ | 300 |
| $\rho_0$ | ${}^{\times}\delta\boldsymbol{x} \sim U(0.9, 1.1)$ ${}^{+}\delta\overline{\boldsymbol{x}}_{x,y} \sim U(0.1), {}^{+}\delta\overline{\boldsymbol{x}}_z \sim U(0, 0.05)$ |
| $P$ | MPM |
| $h$ | $S \cdot 5\mathrm{e}{-5}$ |
| $S$ | 40 |
| $\mu_b$ | 0.9 |
| $N_g$ | $48^3$ |
| $N_{\boldsymbol{x}}$ | 2592 |
| $\rho$ | 1.0 |
| $E$ | 5000. |
| $\nu$ | 0.2 |
| $\sigma_y$ | 50. |

*Table 9:* **RollingFlat task definition**. Based on RollingPin from (Huang et al., 2020).

The derived quantities correspond to the rolling pin's CoM. We use target flatten height $h_{flat} = 0.125$. For the reward function, the first term $R_d$ minimizes the difference between the particles' CoM and $h_{flat}$, while the second term maximizes the smoothness of the particles $R_{flat}$ :

$$d = \frac{\overline{\boldsymbol{x}'}_z}{h_{flat}}$$
$$R_d = \left(\frac{1}{1+d}\right)^2 [\mathbb{1}\{d > 0.33\} \cdot 1 + \mathbb{1}\{d \le 0.33\} \cdot 2]$$
$$R_{flat} = -\mathrm{Var}[\boldsymbol{x}'_z]$$

Initial volume and CoM transform of the particles are randomized.

### E.1.4 SOFTJUMPER

| | |
|---|---|
| $\mathcal{O}$ | $[\mathbb{R}^{204 \times 3}, \mathbb{R}^3, \mathbb{R}^3, \mathbb{R}^{222}] : [\boldsymbol{x}', \overline{\boldsymbol{x}'}, \overline{\boldsymbol{v}'}, \boldsymbol{a}_{t-1}]$ |
| $\mathcal{A}$ | $\mathbb{R}^{222}$ : absolute tetrahedral volumetric activations (subsampled) |
| $R$ | $\overline{\boldsymbol{v}'}_x + (3.0 R_{up} - 0.0001 \sum \|\boldsymbol{a}\|_2^2)$ |
| $d$ | - |
| $T$ | 300 |
| $\rho_0$ | $^\times \delta \boldsymbol{x} \sim U(0.9, 1.1)$ 
 $^+\delta \overline{\boldsymbol{x}}_{x,y} \sim U(0.8), {}^+\delta \overline{\boldsymbol{x}}_z \sim U(0, 0.4)$ |
| $P$ | FEM |
| $h$ | $1/60$ |
| $S$ | 80 |
| $N_{\boldsymbol{x}}$ | 204 |
| $N_{tet}$ | 444 |
| $\rho$ | 1.0 |
| $\lambda$ | 1000. |
| $\mu$ | 1000. |
| $k_{damp}$ | 1.0 |

*Table 10:* **SoftJumper task definition**. Inspired by (Murthy et al., 2021; Hu et al., 2020).

Denote the default initial height $h_0 = \overline{\boldsymbol{x}}_z = 0.2$. The reward function maximizes the forward velocity $\overline{\boldsymbol{v}'}_x$ of the quadruped's CoM, with two auxiliary reward terms $R_{up} = \overline{\boldsymbol{x}'}_z - h_0$ to encourage jumping and the other to encourage energy-efficient policies by minimizing action norms. For the action space, we downsample by a factor of 2 and upsample actions to obtain the full set of activations. Initial volume and CoM transform of the quadruped are randomized.

### E.1.5  HANDFLIP

| | |
|---|---|
| $\mathcal{O}$ | $[\mathbb{R}^{250 \times 3}, \mathbb{R}^3, \mathbb{R}^{24}] : [\boldsymbol{x}', \overline{\boldsymbol{x}'}, q]$ |
| $\mathcal{A}$ | $\mathbb{R}^{24}$ : relative joint angles |
| $R$ | $NI\{R_{flip}\}$ |
| $d$ | - |
| $T$ | 300 |
| $\rho_0$ | $^{\times}\delta\boldsymbol{x} \sim U(0.95, 1.05)$ |
| | $^{+}\delta\overline{\boldsymbol{x}}_{x,y} \sim U(0.1), ^{+}\delta\overline{\boldsymbol{x}}_z \sim U(0, 0.1)$ |
| $P$ | MPM |
| $h$ | $S \cdot 5\text{e}{-}5$ |
| $S$ | 40 |
| $\mu_b$ | 0.5 |
| $N_g$ | $48^3$ |
| $N_{\boldsymbol{x}}$ | 2500 |
| $\rho$ | 1.0 |
| $E$ | 4000. |
| $\nu$ | 0.2 |
| $\sigma_y$ | 130. |

*Table 11:* **HandFlip task definition**. Based on Flip from (Li et al., 2023a).

We define $NI\{r\} = \text{clamp}(\frac{r_0 - r}{r_0}, -1.0, 1.0)$ as the normalized improvement of $r$ from initial $r_0$. The reward function maximizes the $NI$ of $R_{flip}$, where the term $R_{flip} = \overline{\|[\boldsymbol{x}']_{h_l} - [\boldsymbol{x}']_{h_u}\|_2}$ minimizes the average Euclidean distance between two mirrored halves $h_l, h_u$ of the particles. Initial volume and CoM transform of the particles are randomized.

### E.1.6 FLUIDMOVE

| | |
|---|---|
| $\mathcal{O}$ | $[\mathbb{R}^{250\times3}, \mathbb{R}^3, \mathbb{R}^3, \mathbb{R}^3] : [\boldsymbol{x}', \overline{\boldsymbol{x}'}, p, p^*]$ |
| $\mathcal{A}$ | $\mathbb{R}^3$ : relative positions $p$ |
| $R$ | $R_d + R_{spill}$ |
| $d$ | - |
| $T$ | 300 |
| $\rho_0$ | - |
| $\rho^*$ | $p^* \leftarrow [{}^+\delta p_{x,y} \sim U(0.2), {}^+\delta p_z \sim U(0.0, 0.3)]$ |
| $P$ | MPM |
| $h$ | $S \cdot 5e{-}4$ |
| $S$ | 40 |
| $\mu_b$ | 0.5 |
| $N_g$ | $48^3$ |
| $N_{\boldsymbol{x}}$ | 2880 |
| $\rho$ | $1e3$ |
| $E$ | $1e5$ |
| $\nu$ | 0.3 |

*Table 12:* **FluidMove task definition**. Based on TransportWater from (Lin et al., 2021).

For the reward function, the first term $R_d$ minimizes the distance to the target position $p^*$, while the second term penalizes particles which are spilled outside the container :

$$d = \|p - p^*\|_2^2$$
$$R_d = \left(\frac{1}{1+d}\right)^2 [\mathbb{1}\{d > 0.02\} \cdot 1 + \mathbb{1}\{d \leq 0.02\} \cdot 2]$$
$$R_{spill} = \text{see (Lin et al., 2021)}$$

Target position for the container is randomized.

## F    EXTENDED EXPERIMENTAL RESULTS

### F.1    EXAMPLE OF SHAC GETTING STUCK IN LOCAL OPTIMA

We reproduce the original DFlex Ant results from SHAC (Xu et al., 2021), and in Figure 5 we visualize individual runs for insight (Patterson et al., 2023). From this, we observe that one of the runs quickly plateaus to a suboptimal policy after 1M steps and does not improve.

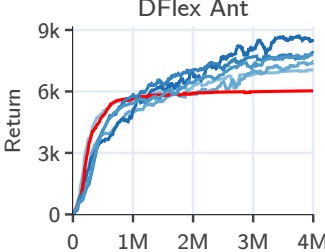

*Figure 5:* **Example of SHAC getting stuck in local minima.** Episode return as a function of environment steps in DFlex Ant ($\mathcal{A} \subset \mathbb{R}^8$). One run (colored in red) quickly plateaus after 1M steps and does not improve. 6 random seeds.

### F.2    RESULTS ON DFLEX LOCOMOTION

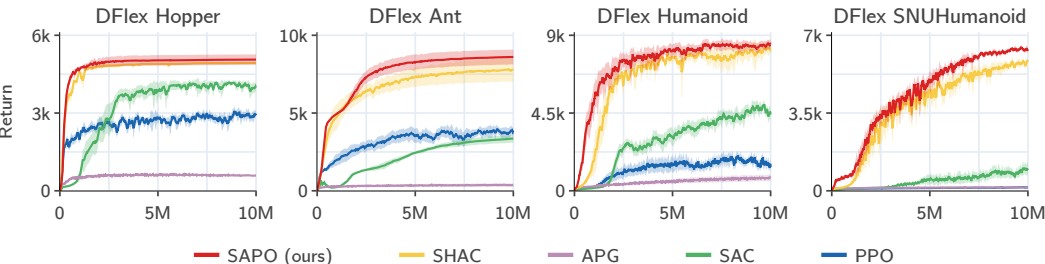

*Figure 6:* **DFlex locomotion training curves.** Episode return as a function of environment steps in DFlex Hopper ($\mathcal{A} \subset \mathbb{R}^3$), Ant ($\mathcal{A} \subset \mathbb{R}^8$), Humanoid ($\mathcal{A} \subset \mathbb{R}^{21}$), and SNUHumanoid ($\mathcal{A} \subset \mathbb{R}^{152}$) locomotion tasks. Mean and 95% CIs over 10 random seeds.

|  | **Hopper** | **Ant** | **Humanoid** | **SNUHumanoid** |
|---|---|---|---|---|
| PPO | $3155 \pm 30$ | $3883 \pm 60$ | $414 \pm 45$ | $135 \pm 3$ |
| SAC | $3833 \pm 50$ | $3366 \pm 25$ | $4628 \pm 120$ | $846 \pm 44$ |
| APG | $590 \pm 3$ | $368 \pm 11$ | $783 \pm 16$ | $149 \pm 1$ |
| SHAC | $4939 \pm 3$ | $7779 \pm 70$ | $8256 \pm 74$ | $5755 \pm 67$ |
| SAPO (ours) | $5060 \pm 18$ | $8610 \pm 40$ | $8469 \pm 58$ | $6427 \pm 53$ |

*Table 13:* **DFlex locomotion tabular results.** Evaluation episode returns for final policies after training. Mean and 95% CIs over 10 random seeds with 128 episodes per seed.

### F.3 SAPO ABLATIONS ON DFLEX LOCOMOTION

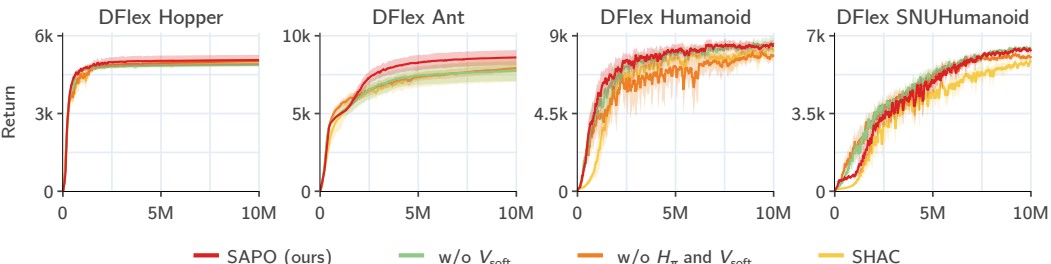

*Figure 7:* **SAPO ablations – DFlex locomotion training curves.** Episode return as a function of environment steps in DFlex Hopper ($\mathcal{A} \subset \mathbb{R}^3$), Ant ($\mathcal{A} \subset \mathbb{R}^8$), Humanoid ($\mathcal{A} \subset \mathbb{R}^{21}$), and SNUHumanoid ($\mathcal{A} \subset \mathbb{R}^{152}$) locomotion tasks. Mean and 95% CIs over 10 random seeds.

|  | **Hopper** | **Ant** | **Humanoid** | **SNUHumanoid** | **(avg $\Delta\%$)** |
|---|---|---|---|---|---|
| SAPO (ours) | $5060 \pm 18$ | $8610 \pm 42$ | $8469 \pm 59$ | $6427 \pm 52$ | 6.8% |
| w/o $V_{\text{soft}}$ | $4882 \pm 7$ | $7729 \pm 52$ | $8389 \pm 76$ | $6392 \pm 54$ | 2.7% |
| w/o $\mathcal{H}_\pi$ and $V_{\text{soft}}$ | $5036 \pm 2$ | $7897 \pm 30$ | $7731 \pm 91$ | $6032 \pm 58$ | 0.5% |
| SHAC | $4939 \pm 3$ | $7779 \pm 69$ | $8256 \pm 76$ | $5755 \pm 66$ | – |

*Table 14:* **SAPO ablations – DFlex locomotion tabular results.** Evaluation episode returns for final policies after training. Mean and 95% CIs over 10 random seeds with 128 episodes per seed.

### F.4 ADDITIONAL SAPO ABLATIONS FOR DESIGN CHOICES {III, IV, V}

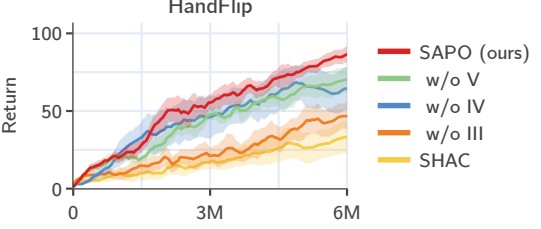

|  | **HandFlip** | **($\Delta\%$)** |
|---|---|---|
| SAPO (ours) | $90 \pm 2$ | +172.7% |
| w/o V | $71 \pm 3$ | +115.2% |
| w/o IV | $64 \pm 3$ | +93.9% |
| w/o III | $49 \pm 3$ | +48.5% |
| SHAC | $33 \pm 3$ | – |

*Figure 8:* **SAPO ablations – (ext.) HandFlip training curves.** Episode return as a function of environment steps. Smoothed using EWMA with $\alpha = 0.99$. Mean and 95% CIs over 10 random seeds.

*Table 15:* **SAPO ablations – (ext.) HandFlip tabular results.** Evaluation episode returns for final policies after training. Mean and 95% CIs over 10 random seeds with 64 episodes per seed.

See Section 4.2 for descriptions of these design choices.

### F.5 RUNTIME AND SCALABILITY OF REWARPED

We report all timings on a consumer workstation with an AMD Threadripper 5955WX CPU, NVIDIA RTX 4090 GPU, and 128GB DDR4 3200MHz RAM. We test the scalability of Rewarped's parallel MPM implementation over an increasing number of environments, on the HandFlip task. See Appendix E.1.5 for details on environment settings.

We run 5 full-length trajectories, and report the averaged runtime of forward simulation. As a reference, the non-parallel MPM implementation from (Li et al., 2023a) runs with a single environment at 70 FPS using 3.51 GB of GPU memory (when $N_x = 2500$, $N_g = 48^3$).

With one environment, Rewarped's MPM implementation is $3\times$ faster and uses $2/3$ less memory compared to DexDeform's. Using 32 environments, Rewarped achieves a $20\times$ total speedup compared to DexDeform for simulating the HandFlip task. Furthermore, the GPU memory used *per env* decreases because overhead is amortized across multiple environments that are simulated in same underlying physics scene. In Figure 9, we observe that the total FPS drastically increases until plateauing beyond $N = 32$ environments. This occurs because GPU utilization is maxed out; more GPUs or a GPU with more CUDA cores would be needed to continue scaling.

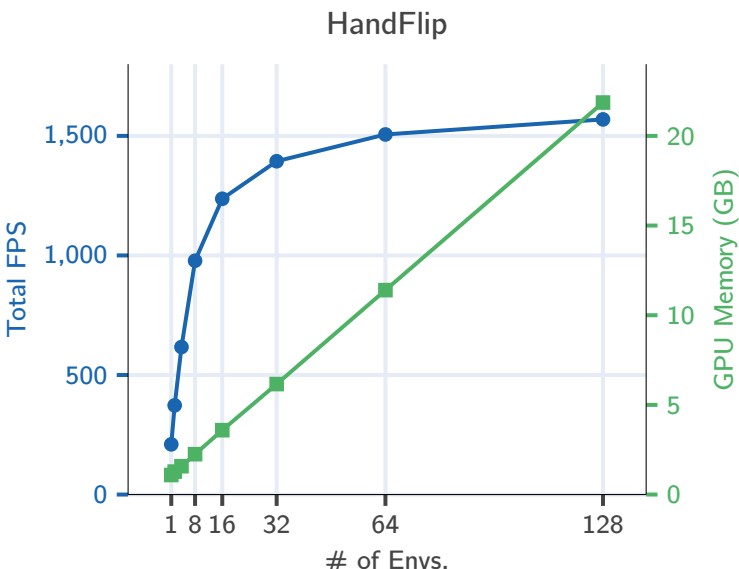

*Figure 9:* **Rewarped runtimes – HandFlip curves.** Total FPS (blue) and GPU memory (green) are shown. Total FPS averaged over $5N$ full-length trajectories with random actions.

| # of Envs. | Total FPS | Relative Speedup | GPU Memory (GB) |
|---|---|---|---|
| 1 | 210 | 1.00 | 1.09 |
| 2 | 373 | 1.78 | 1.28 |
| 4 | 617 | 2.94 | 1.58 |
| 8 | 978 | 4.66 | 2.25 |
| 16 | 1,237 | 5.89 | 3.59 |
| 32 | 1,394 | 6.64 | 6.16 |
| 64 | 1,506 | 7.17 | 11.40 |
| 128 | 1,569 | 7.47 | 21.86 |

*Table 16:* **Rewarped runtimes – HandFlip tabular results.** Total FPS averaged over $5N$ full-length trajectories with random actions.

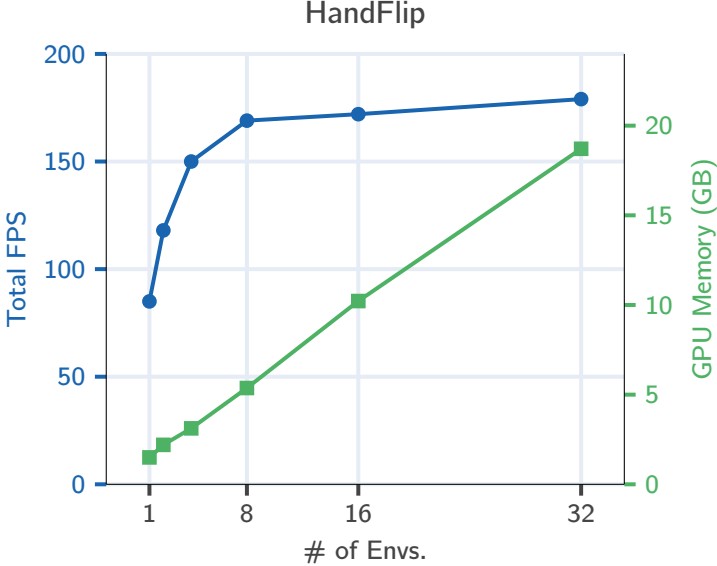

*Figure 10:* **Rewarped runtimes – HandFlip** ($N_x = 30000$) **curves.** Total FPS (blue) and GPU memory (green) are shown. Total FPS averaged over $5N$ full-length trajectories with random actions.

| # of Envs. | Total FPS | Relative Speedup | GPU Memory (GB) |
|:---:|:---:|:---:|:---:|
| 1 | 85 | 1.00 | 1.50 |
| 2 | 118 | 1.39 | 2.20 |
| 4 | 150 | 1.76 | 3.12 |
| 8 | 169 | 1.99 | 5.37 |
| 16 | 172 | 2.02 | 10.22 |
| 32 | 179 | 2.11 | 18.71 |

*Table 17:* **Rewarped runtimes – HandFlip** ($N_x = 30000$) **tabular results.** Total FPS averaged over $5N$ full-length trajectories with random actions.

In Table 18, we report training runtimes for algorithms on the HandFlip task, averaged over 10 random seeds. We also note that runtimes may be further optimized using PyTorch compile and CUDA graphs, but we did not implement this.

| Algorithm | Runtime (hours) |
|:---|:---:|
| PPO | 1.46 |
| SAC | 1.67 |
| APG | 7.43 |
| SHAC | 7.66 |
| SAPO (ours) | 7.77 |

*Table 18:* **Training runtimes – HandFlip task.** Algorithm training runtimes over 6M environment steps. Mean over 10 random seeds.

## F.6 Visualizations of trajectories in Rewarped tasks

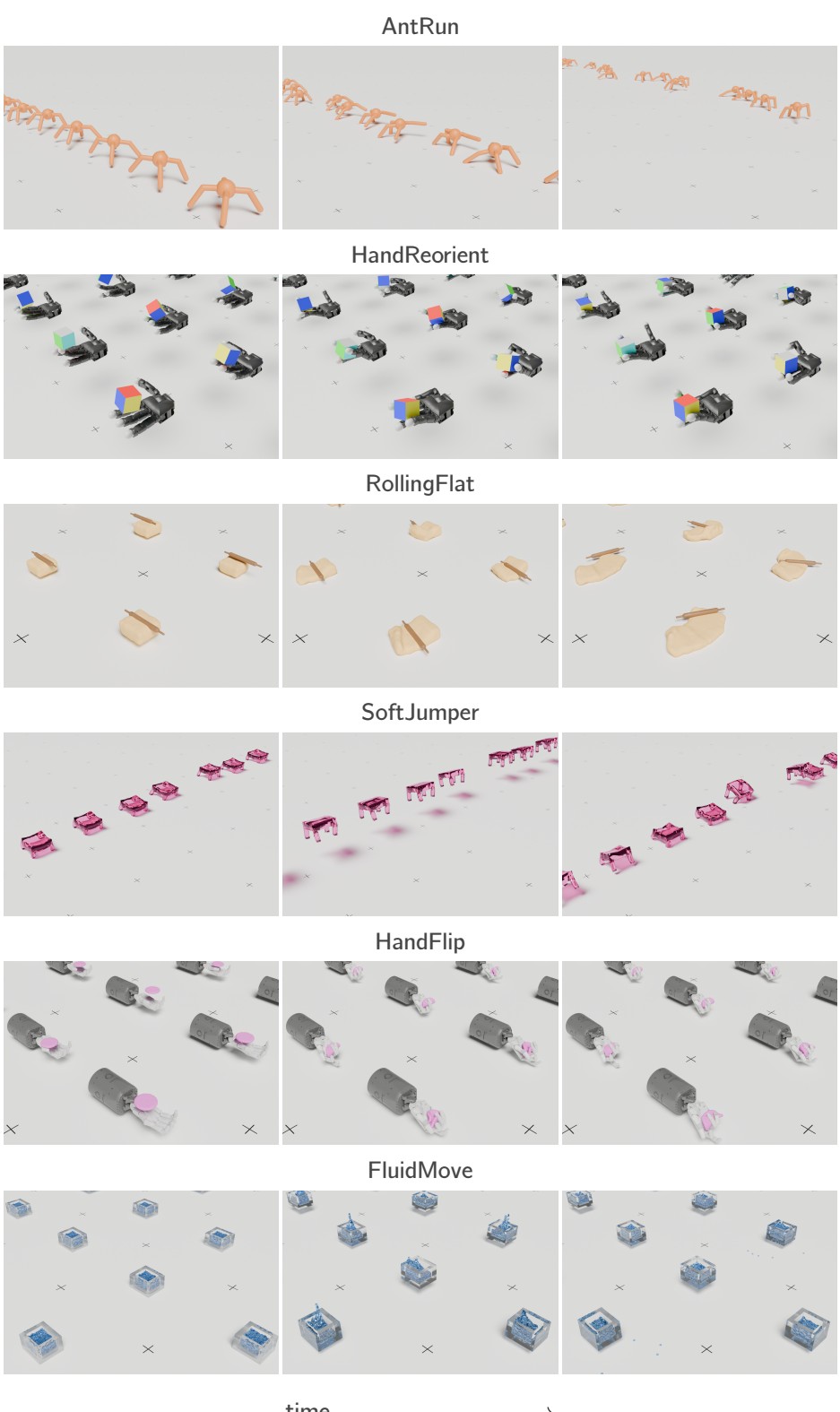

*Figure 11:* **Visualizations of trajectories from policies learned by SAPO in Rewarped tasks.** The camera view is fixed between different time steps.

## F.7 VISUALIZATIONS OF OPTIMIZATION LANDSCAPE

Following (Xu et al., 2021), we visualize loss surfaces to compare different algorithms and analyze the smoothness of the optimization landscape. We use undiscounted episodic returns for the loss values, and apply a symlog scale for visualization. We sample two filter-normalized random directions in policy parameter space and evaluate perturbations to final policies after training. We compare algorithms trained on DFlex Ant (Figure 13) and Rewarped HandFlip (Figure 12), observing that loss surfaces for SAPO appear to be flatter and smoother, compared to those of APG or SHAC.

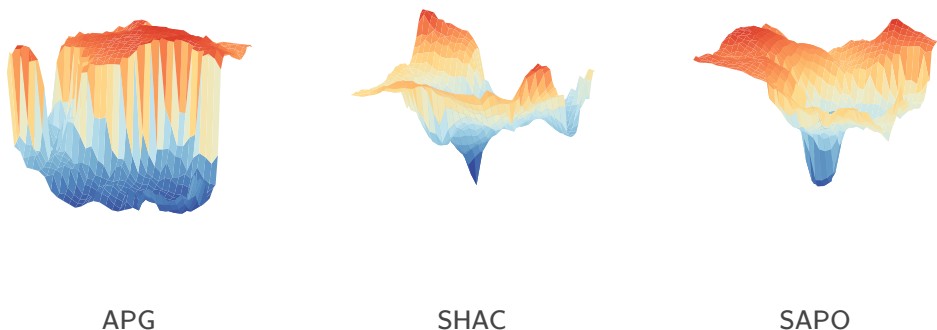

APG          SHAC          SAPO

*Figure 12:* **Loss surface comparison between algorithms – DFlex Ant.**

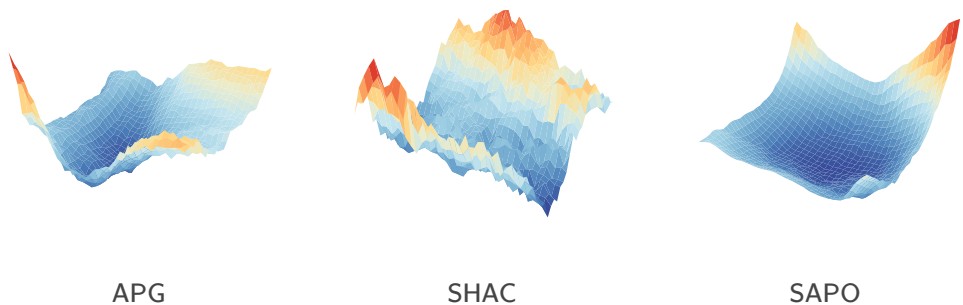

APG          SHAC          SAPO

*Figure 13:* **Loss surface comparison between algorithms – Rewarped HandFlip.**

