# OpenReview forum: "Stabilizing Reinforcement Learning in Differentiable Multiphysics Simulation"
_ICLR.cc/2025/Conference — ICLR 2025 Spotlight_

### Official Review · Reviewer_cLkG · 2024-10-26

**Soundness:** 3
**Presentation:** 4
**Contribution:** 3
**Rating:** 8
**Confidence:** 4

**Summary:**

This work proposes a first-order model-based reinforcement learning (FO-MBRL) method that combines differentiable simulation and entropy maximization. It also provides a new parallel differentiable simulator that supports various physical systems. The paper demonstrates the efficacy of its proposed FO-MBRL on typical test environments implemented in the new differentiable simulator and reports its performance gain over several representative baselines from reinforcement learning and differentiable simulation.

**Strengths:**

- The paper is well-organized and nicely written. In particular, I like the way the paper organizes the background section and the preamble in the experiment section;
-  The discussion on the design decisions (Sec. 4.2) is informative for practitioners. The paper is upfront with these specific choices in the framework, which I really appreciate;
- The quality of the simulation environments seems respectable, and the rendering is well-polished (Fig. 1). I am inclined to believe this differentiable simulator has the potential to be a major player in the future, but I’d like to check a few more details (See weaknesses).
- The experimental results and the ablation study look convincing to me.

**Weaknesses:**

One contribution claimed by this work is the new differentiable simulator that simultaneously accommodates GPU parallelization, multiphysics, and differentiability.
- I didn’t find detailed specifications of the environments, or maybe I missed them. The claim of this simulator is quite ambitious, and verifying this claim requires scrutinizing many technical details. In particular, I am looking for at least the following information: 1) the simulation problem size, including grid and particle numbers in MPM and the time step sizes; 2) the treatment of multiphysics coupling and their gradients, e.g., rigid-soft contact and solid-fluid coupling as shown in Fig. 1.
- Table 1 and the related work should also discuss DiffTaichi, which I believe supports GPU parallelization, deformable solids and fluids, and gradient computation.
- While I highly appreciate the engineering effort behind Rewarped, I don’t see much technical/algorithmic novelty in Sec. 5: the core simulation algorithms behind rigid/soft/fluids are from existing (differentiable-)MPM papers and GPU parallelization is offered by Warp. Grinding existing differentiable simulators in Table 1 and the related work section is a bit unnecessary :)

**Questions:**

I am generally happy with this work, and my major concern has been listed in the weakness section. Several very minor questions:
- I am under the impression that there should have been a log function in Eqn. 4 (somewhat similar to Eqn. 2 in Georgiev 24). Could you confirm your Eqn. 4 is correct?
- Eqn 12: where is (sl, al) in the inner expectation sampled from? Does this notation miss a probability distribution?
- How is your simulation gradient backprogagated through the stochastic policy tanh (line 332) exactly?

---

> ### Author Response · Authors · 2024-11-22
> **Authors' Response to Reviewer cLkG [1/1]**
>
> We thank the reviewer for their time and kind words regarding our work on Rewarped and SAPO, as well as on the presentation of the paper. We proceed to address each of the reviewer’s comments and questions.
>
> - - -
>
> > [W1] … “detailed specifications of the environments” …
>
> We added **two new sections** to the appendix, which should resolve the reviewer’s questions. Appendix D discusses the underlying physics-based simulation methods used in Rewarped. Appendix E describes the implementation details for every task, e.g. relevant physics hyperparameters (like grid size, num particles, time step), observation/action/reward definitions.
>
> > [W2] Table 1 and the related work should also discuss DiffTaichi, which I believe supports GPU parallelization, deformable solids and fluids, and gradient computation.
>
> We use Table 1 and Section 2 to review *parallel* (differentiable) simulators that are capable of running many environments together, like in [[IsaacGym](https://arxiv.org/abs/2108.10470)]. To the best of our knowledge, DiffTaichi focuses on GPU parallelization for efficient kernel computation (ie. parallelizing operations across the MPM grid) **in a single environment instance**. These prior works [[DiffTaichi](https://arxiv.org/abs/1910.00935), [PlasticineLab](https://arxiv.org/abs/2104.03311), [DexDeform](https://arxiv.org/abs/2304.03223), [FluidLab](https://arxiv.org/abs/2303.02346)] do not support parallel simulation natively.
>
> We have also added a **new section** (Appendix A) that extends discussion of related work to non-parallel differentiable simulators, including DiffTaichi.
>
> > [W3] “While I highly appreciate the engineering effort behind Rewarped, I don’t see much technical/algorithmic novelty in Sec. 5” …
>
> The main contribution within Section 5 is the introduction of our simulation platform that enables scaling RL on tasks involving rigid and soft body interaction, such as SAPO, the novel FO-MBRL algorithm we propose in Section 4.
>
> Absolutely, Rewarped would not be possible without the amazing work that has already been done by the physics-based simulation, graphics, ML, RL, or robotics communities. However, to the best of our knowledge, no existing simulation platform satisfies all these desiderata--parallel environments, differentiability, multiphysics (for interaction between rigid bodies and deformables). Rewarped aims to close this gap. Table 1 and the related work section on parallel differentiable simulation contextualizes Rewarped to explain these desiderata and motivation behind why we built Rewarped, rather than using an existing solution (it did not exist). We believe that readers will also benefit from these comparisons. Rewarped is a major contribution of this paper and without it, we would not be able to easily demonstrate SAPO across a variety of tasks.
>
> To further clarify, while Warp supports kernel-level GPU parallelization, we made substantial effort developing Rewarped to support (a) parallel simulation of multiple environments (as shown in Figure 1), as well as the other two criteria: (b) efficient batched computation of simulation gradients, for use with analytic policy optimization in a scalable RL pipeline; (c) multiphysics support for a variety of different material types. We have tried to make this clearer within Section 5, by now referring to “parallel simulation” for many environments vs. “non-parallel simulation” for a single environment.
>
> We hope the research community will appreciate our efforts in building Rewarped, and that Rewarped can enable future research on learning with parallel differentiable simulation or scaling RL for rigid & soft body tasks.
>
> > [Q1] I am under the impression that there should have been a log function in Eqn. 4 (somewhat similar to Eqn. 2 in Georgiev 24). Could you confirm your Eqn. 4 is correct?
>
> Thanks for catching this typo! Yes, it should be $\nabla_{\theta} \log \pi$, we have updated the paper to correct this.
>
> > [Q2] Eqn 12: where is (sl, al) in the inner expectation sampled from? Does this notation miss a probability distribution?
>
> Yes, we omitted the distribution for the inner expectation, and have corrected this. Thanks for pointing this out!
>
> > [Q3] How is your simulation gradient backprogagated through the stochastic policy tanh (line 332) exactly?
>
> We do this by applying the reparameterization trick, as done in SAC [[Haarnoja et al., 2018]](https://arxiv.org/abs/1812.05905).
>
> - - -
>
> **Overall**, we think these changes & clarifications should resolve all of the reviewer’s concerns. We hope that the reviewer will decide to raise their confidence and scores accordingly. If there are any additional questions, we are happy to discuss further!

---

> > ### Comment · Reviewer_cLkG · 2024-11-25
> > **Thank you for the response**
> >
> > Thank you for the clarification.
> >
> > W1: thank you for adding the new sections in Appendix that detail the simulation environment setup. Looking at the statistics in Appendix E, it looks like the simulation instances typically consist of < 3000 particles (reading from the N_x number in the tables). For a 3D deformable-body/fluid scene, this is a pretty small size and 10x smaller than what DiffTaichi demonstrated (Fig. 6 in Sec. E1 of DiffTaichi reported a 30K-particle scene). I suggest the main paper remind readers of this difference in particle numbers.
> >
> > W2: based on W1, my feeling is that if only one GPU is available, the implementation in this paper is better at parallelizing multiple small-sized simulation problem instances whereas DiffTaichi is better suited for parallelizing the computation in simulating one large scene. It will be great if the authors can confirm or correct my understanding. Is there an "upper-limit" of particle/grid numbers in one simulation scene that makes your simulator struggle when parallelizing multiple instances of such scenes?

---

> > > ### Author Response · Authors · 2024-11-25
> > > **Additional scaling results on scenes with more particles**
> > >
> > > > W2: based on W1, my feeling is that if only one GPU is available, the implementation in this paper is better at parallelizing multiple small-sized simulation problem instances whereas DiffTaichi is better suited for parallelizing the computation in simulating one large scene. It will be great if the authors can confirm or correct my understanding. Is there an "upper-limit" of particle/grid numbers in one simulation scene that makes your simulator struggle when parallelizing multiple instances of such scenes?
> > >
> > > (Aside: These settings can be adjusted depending on the end-user’s use cases. In this work, we focused on parallel differentiable simulation, so we used $N_x = 2500$ per env, which corresponds to 80000 total particles over 32 environments.)
> > >
> > > Here are new scaling results simulating the HandFlip task with $N_x = 30000$. We can forward simulate **960000 total particles at ~180 total FPS, using 30000 particles per env * 32 envs**. For DexDeform, simulating *one* environment using $N_x=30000$ runs at 26.55 FPS using 10.97 GB of GPU memory (they do not use gradient checkpointing).
> > >
> > > | # of Envs. | Total FPS | GPU Memory (GB) |
> > > | :- | :-: | :-: |
> > > | 1 | 85 | 1.50 |
> > > | 2 | 118 | 2.20 |
> > > | 4 | 150 | 3.12 |
> > > | 8 | 169 | 5.37 |
> > > | 16 | 172 | 10.22 |
> > > | 32 | 179 | 18.71 |
> > >
> > > We did not use this setting for the main results in the paper, as it would slow down parallel simulation and decrease available memory for training RL, given the limited compute budget we have access to.
> > >
> > > We would like to point out that prior work [[RoboCraft](https://arxiv.org/abs/2205.02909), [DP3](https://arxiv.org/abs/2403.03954)] has shown that models that take as observations ~300 particles subsampled from MPM simulation can transfer to the real world and perform complex deformable object manipulation tasks. In particular, simulation settings for robotic manipulation tasks (the domain we aim for in this work) may differ from high-fidelity simulation for computer graphics (which DiffTaichi targets).
> > >
> > > Regarding upper limits, we note that larger scenes and more environments can be simulated using more powerful GPUs (with more CUDA cores and available memory). As of now, we only have access to RTX 4090s which have 24 GB of GPU memory. We can share scaling results with more powerful GPUs in camera ready if the reviewer deems it necessary.
> > >
> > > > W1: thank you for adding the new sections in Appendix that detail the simulation environment setup. Looking at the statistics in Appendix E, it looks like the simulation instances typically consist of < 3000 particles (reading from the N_x number in the tables). For a 3D deformable-body/fluid scene, this is a pretty small size and 10x smaller than what DiffTaichi demonstrated (Fig. 6 in Sec. E1 of DiffTaichi reported a 30K-particle scene). I suggest the main paper remind readers of this difference in particle numbers.
> > >
> > > Per the reviewer’s request, we have added a comment on the number of simulation particles to Section 6 (L417). Note that the scales we use are more similar to [SoftGym, PlasticineLab, DexDeform], which use particle-based simulation / MPM for a range of sequential decision making tasks, while DiffTaichi is focused on simulation itself. We keep particle sizes similar across tasks in Rewarped for fairer comparison between tasks.
> > >
> > > - - -
> > >
> > > **Overall**, we believe these results should further resolve the reviewer’s concerns, and hope that the reviewer will increase their score accordingly.

---

> > > > ### Comment · Reviewer_cLkG · 2024-11-26
> > > > **Thank you for the new result**
> > > >
> > > > I appreciate the new results, and I am now more confident with this work. I will update my confidence score (3 -> 4) to show my support for your work.

---

### Official Review · Reviewer_W9i4 · 2024-11-04

**Soundness:** 3
**Presentation:** 3
**Contribution:** 3
**Rating:** 8
**Confidence:** 3

**Summary:**

This paper releases a new algorithm called SAPO which acts better on Soft rigid body dynamics and also a simulator, with the main contribution being sample efficient RL on these kinds of problems. SAPO is a model-based, actor-critic method that leverages analytic gradients from a differentiable simulator to optimize a stochastic policy with entropy maximization.  Rewarped is the simulator, a  differentiable multi-physics simulation platform capable of simulating soft bodies which usually have slow simulation times. They demonstrate that SAPO with Rewarped achieves superior performance on manipulation and locomotion tasks that require handling interactions between rigid structures and deformables outperforming baseline approaches.

**Strengths:**

1. The paper is well motivated. The authors take inspiration from max entropy model free RL and extend this to SOTA first order model based RL.
2. The contribution of a accelerated differentiable simulator for soft body dynamics is welcome.
3. The paper is comprehensive in providing an ablation over components of their method and also provides explanations for their design choices.

**Weaknesses:**

1. Since this work is inspired by model-based reinforcement learning (MBRL) and maximum entropy RL, and the authors present a first-order MBRL algorithm based on the max-entropy RL framework, they should clearly distinguish their approach from prior work, such as this paper(https://arxiv.org/abs/2112.01195) and this one (https://ieeexplore.ieee.org/document/9892381). While it can be argued that these works use a world model rather than a differentiable simulation, this distinction should be explicitly discussed. A brief discussion on the potential inter-applicability of world model and gradient simulator-based methods could open up promising directions for future research. A comparison of the novelty, applications, and results of these works would strengthen the contribution of the paper.
2. It is not directly intuitive as to why the review of Zeroth Order Batched Gradient is necessary in the main manuscript in Section 3 Equation 4. It is my understanding that the paper builds on top of first order model based RL methods instead. If this section is indeed required please elaborate on it's importance.

**Questions:**

1. How does the simulation speed scale with number of particles in the soft body simulation ? A discussion on this can greatly help in understanding the applicability of Rewarped. A potential experiment of interest can be simulation speed vs number of particles.
2. Similarly, what is the effect of number of parallel environments on the simulation speed ? Similar to the above experiment, a table/plot showing simulation speed vs num of parallel environments on a standard consumer grade GPU (eg. 4090)

---

> ### Author Response · Authors · 2024-11-22
> **Authors' Response for Reviewer W9i4 [1/2]**
>
> We thank the reviewer for their time, and positive comments on the strengths of SAPO and Rewarped. Based on the reviewers’ feedback, we have updated the paper to include results on the runtime performance / scalability of Rewarped and extended the discussion of related work.
>
> - - -
>
> > [Q1] How does the simulation speed scale with number of particles in the soft body simulation ? A discussion on this can greatly help in understanding the applicability of Rewarped. A potential experiment of interest can be simulation speed vs number of particles.
>
> and
>
> > [Q2] Similarly, what is the effect of number of parallel environments on the simulation speed ? Similar to the above experiment, a table/plot showing simulation speed vs num of parallel environments on a standard consumer grade GPU (eg. 4090)
>
> We have added a **new subsection** (Appendix F.5) with results on the runtime performance and scalability of Rewarped.
>
> We achieve 20x total speedup with Rewarped compared to DexDeform, using a single RTX 4090 GPU:
> - Rewarped: 32 envs, ~1400 FPS total, 80000 particles total
> - DexDeform (non-parallel simulator): 1 env, ~70 FPS, 2500 particles
> - Rewarped: 1 env, ~210 FPS, 2500 particles
>
> Regarding GPU memory usage:
> - Rewarped: 0.19 GB per env, 6.16 GB total over 32 envs
> - DexDeform: 3.51 GB over 1 env
>
> Further results are available in Appendix F.5.
>
> … (continue in response 2/2)

---

> ### Author Response · Authors · 2024-11-22
> **Authors' Response for Reviewer W9i4 [2/2]**
>
> …
>
> > [W1] … “clearly distinguish their approach from prior work” …
>
> To better situate SAPO against prior literature in model-based RL, we added a **new section** in Appendix A to extend discussion of related work. We also note that our list of references is already very long for a paper that is not a literature review. We have tried our best to include relevant literature spanned across multiple fields, given the nature of this work.
>
> We have added discussion of SAC-SVG [[Amos et al., 2021](https://arxiv.org/abs/2008.12775)] to Section 4.1, which we found was the earliest & closest comparison for SAPO from the model-based RL literature. SAC-SVG incorporates entropy regularization, soft $Q$-value estimates, and short-horizon stochastic value gradients using a learned deterministic world model. As the reviewer points out, our work uses analytic policy gradients from differentiable simulation instead of backpropagating through a learned world model. We also discuss the challenges of world modeling approaches for deformable tasks in Section 1: Introduction, to motivate why we use a FO-MBRL approach in our work, instead of learning to predict dynamics.
>
> For [[Svidchenko and Shpilman, 2021](https://arxiv.org/abs/2112.01195)], based on our understanding, they combine state entropy-based exploration [[Hazan et al., 2019](https://arxiv.org/abs/1812.02690)] with Dreamer [[Hafner et al., 2020](https://arxiv.org/abs/1912.01603)], which explicitly learns a world model, and only evaluate on rigid-body locomotion tasks. By contrast, SAPO considers policy entropy regularization and learns locomotion & manipulation tasks involving rigid & soft body interaction.
>
> Whereas [[Ma et al., 2022](https://ieeexplore.ieee.org/document/9892381)] incorporate policy entropy regularization and soft $Q$-values into Dreamer (compared to SAC-SVG which does the same for SAC), they also only evaluate on rigid-body continuous control tasks. From [Algorithm 1 & Section IV of Ma et al., 2022], it is unclear to us whether they apply automatic temperature tuning like in SAC, SAC-SVG, or SAPO (ours). Furthermore, we introduce several design choices in Section 4.2, such as eliminating target networks, to further improve stable training of SAPO (also see **new subsection** Appendix F.4 for further ablations). In comparison, [Ma et al., 2022] require target networks for both the critic and actor to train their world model.
>
> In summary, SAPO is a novel FO-MBRL algorithm which does maximum entropy RL by analytic policy optimization using gradients from differentiable simulation. As the reviewer points out, one promising area to consider for future directions is whether differentiable simulation can inform learned world models, and we have added this to Section 7.
>
> > [W2] It is not directly intuitive as to why the review of Zeroth Order Batched Gradient is necessary in the main manuscript in Section 3 Equation 4. It is my understanding that the paper builds on top of first order model based RL methods instead. If this section is indeed required please elaborate on it's importance.
>
> We provide a review of the ZOBG (Equation 4) in Section 3: Background, in order to frame the introduction of the FOBG (Equation 5) used in FO-MBRL which immediately follows it. It is common to use the ZOBG to estimate policy gradients within vanilla RL. The reviewer is correct that SAPO is a FO-MBRL algorithm which uses the FOBG to compute the analytic policy gradient. We believe the discussion of the ZOBG (including how policy gradient algorithms attempt to reduce its variance) is useful to better understand the FOBG through differentiable simulation, as it is not a common paradigm in RL currently.
>
> - - -
>
> **Overall**, we believe these changes and clarifications should resolve the reviewer’s concerns, and hope that the reviewer will increase their score accordingly. We are happy to discuss additional questions regarding our work.

---

> ### Comment · Reviewer_W9i4 · 2024-11-26
>
> Dear authors, thanks for the clarifications with the detailed explanation. I'll be updating the score to make it an accept with a score of 8 :)

---

### Official Review · Reviewer_wHaW · 2024-11-05

**Soundness:** 3
**Presentation:** 3
**Contribution:** 3
**Rating:** 8
**Confidence:** 4

**Summary:**

This paper introduces Rewarped, a differentiable multiphysics simulator, and SAPO (Soft Analytic Policy Optimization), a maximum entropy RL method designed to leverage differentiable simulation for efficient policy learning. SAPO uses Rewarped’s capabilities to compute on-policy, first-order policy gradients, maximizing a maximum entropy RL objective. Built on NVIDIA's Warp, Rewarped includes a parallelized Material Point Method (MPM)-based soft body simulator with two-way coupling, providing an advanced simulation environment that supports a variety of rigid and soft body manipulation tasks. Benchmark results show that SAPO outperforms current state-of-the-art methods across challenging locomotion and manipulation environments.

**Strengths:**

1. The paper is well-written and the ideas are presented clearly, making it accessible and easy to follow.
2. The parallelized soft body simulator in Rewarped is a notable advancement. Integrating soft body manipulation environments within a unified Warp-based framework provides an excellent foundation for further research on differentiable physics and soft body RL.
3.  SAPO's combination of on-policy soft policy learning with an off-policy critic is an effective method for differentiable physics-based RL. This design mitigates the complexity of offline policy gradient evaluation while still benefiting from the simulator’s analytical gradients.
4. SAPO demonstrates strong performance on complex soft body manipulation tasks, such as dexterous hand manipulation, consistently outperforming baseline methods.
5. The authors’ commitment to releasing the dataset and simulation code enhances the reproducibility and accessibility of their work.

**Weaknesses:**

1. While combining maximum entropy with differentiable physics is a promising approach, it may appear incremental, as similar techniques have been explored in prior RL literature.
2. If I understand correctly, only one-way coupling is considered in MPM simulator. It would be nice if two way coupling could be added to allow impact back on the rigid body as in Maniskill2.
3. The paper lacks a discussion of the efficiency gains from the parallelized MPM implementation. Including an analysis of wall-time efficiency or runtime comparisons would provide valuable insight

**Questions:**

1. A comprehensive ablation study would provide valuable insights into the effectiveness of different design choices. For instance, comparing the performance of SAPO with different activation functions (e.g., ELU vs. SiLU).
2. I am curious about the improvements brought by the paralleled MPM. how much acceleration can be achieved with the paralleled MPM, compared with the version in DexDeform? How much GPU memory is required?

---

> ### Author Response · Authors · 2024-11-22
> **Authors' Response for Reviewer wHaW [1/2]**
>
> We thank the reviewer for their time and kind words regarding the notable advances presented by Rewarped and SAPO, as well as on the presentation clarity of the paper. We remain committed to releasing all code after publication for the benefit of the community. We proceed to address each of the reviewer’s comments and questions.
>
> - - -
>
> > [W1] While combining maximum entropy with differentiable physics is a promising approach, it may appear incremental, as similar techniques have been explored in prior RL literature.
>
> To better situate SAPO against prior literature in model-based RL, we added a **new section** in Appendix A that extends discussion of related work.
>
> We respectfully disagree with the reviewer’s perspective. We believe SAPO, which integrates maximum entropy RL with analytic policy gradients from differentiable simulation, is a major novel contribution of this work. Although maximum entropy RL is a topic of interest within vanilla  RL literature, our work is the first formulation of maximum entropy RL using analytic gradients from differentiable simulation for FO-MBRL.
>
> To the best of our knowledge, our work is the first to demonstrate that entropy regularization by maximum entropy RL can stabilize policy optimization over analytic gradients from differentiable simulation. Moreover, this is the first work to show that FO-MBRL can learn a range of locomotion & manipulation tasks involving rigid & soft bodies. Prior work [[Brax](https://arxiv.org/abs/2106.13281), [SHAC](https://arxiv.org/abs/2204.07137)] has only successfully demonstrated FO-MBRL on rigid-body locomotion tasks, and in fact, FO-MBRL previously was shown to have limited or no advantage over model-free RL on deformable tasks [[DaXBench]](https://arxiv.org/abs/2210.13066).
>
> > [W2] If I understand correctly, only one-way coupling is considered in MPM simulator. It would be nice if two way coupling could be added to allow impact back on the rigid body as in Maniskill2.
>
> Yes, we use one-way coupling similar to [[PlasticineLab](https://arxiv.org/abs/2104.03311), [DexDeform](https://arxiv.org/abs/2304.03223)]. Prior work [[RoboCraft](https://arxiv.org/abs/2205.02909), [DP3](https://arxiv.org/abs/2403.03954)] has shown that models trained on one-way coupled MPM simulation data can transfer to the real world, so the utility of two-way coupling may depend on the action parameterization. For instance, position control with one-way coupling may sufficiently model real-world dynamics. Depending on the control frequency, the robot’s underlying motor controllers could compensate for forces applied by soft bodies onto the robot’s rigid end effector. Whereas torque control may require two-way coupling for more accurate dynamics.
>
> We do not believe that this is a major weakness of our work. End-users of Rewarped may also choose to modify the MPM simulation code to incorporate two-way coupling depending on their use cases.
>
> > [W3] The paper lacks a discussion of the efficiency gains from the parallelized MPM implementation. Including an analysis of wall-time efficiency or runtime comparisons would provide valuable insight
>
> We have added a **new subsection** (Appendix F.5) with results on the runtime performance and scalability of Rewarped. Also see comment for [Q2] below.
>
> … (continue in response 2/2)

---

> ### Author Response · Authors · 2024-11-22
> **Authors' Response for Reviewer wHaW [2/2]**
>
> …
>
> > [Q1] A comprehensive ablation study would provide valuable insights into the effectiveness of different design choices. For instance, comparing the performance of SAPO with different activation functions (e.g., ELU vs. SiLU).
>
> We have added a **new subsection** (Appendix F.4) which provides results for individual ablations on design choices {III, IV, V}. Recall these design choices:
>
> - III: state-dependent variance for policy
> - IV: dual critic, no target networks
> - V: SiLU, AdamW
>
> From our experience with early SAPO experiments, (AdamW + SiLU) > (Adam + SiLU) > (Adam + ELU) > (AdamW + ELU), so we opted to combine these modifications in design choice V. If the reviewer thinks it would be crucial, we can include further ablations on the activation function or optimizer alone for camera ready.
>
> From these new ablations, we observe that all three design choices impact the performance of SAPO on the HandFlip task. We find that, ordering in terms of impact on performance, III > IV > V, which agrees with our original motivation regarding these design choices in Section 4.2.
>
> > [Q2] I am curious about the improvements brought by the paralleled MPM. how much acceleration can be achieved with the paralleled MPM, compared with the version in DexDeform? How much GPU memory is required?
>
> We achieve 20x total speedup with Rewarped compared to DexDeform, using a single RTX 4090 GPU:
> - Rewarped: 32 envs, ~1400 FPS total, 80000 particles total
> - DexDeform (non-parallel simulator): 1 env, ~70 FPS, 2500 particles
> - Rewarped: 1 env, ~210 FPS, 2500 particles
>
> Regarding GPU memory usage:
> - Rewarped: 0.19 GB per env, 6.16 GB total over 32 envs
> - DexDeform: 3.51 GB over 1 env
>
> Further results are available in Appendix F.5 (new subsection).
>
> - - -
>
> **Overall**, we believe the changes we’ve made should resolve the reviewer’s remaining concerns, and hope the reviewer may choose to increase their scores accordingly. We are happy to discuss further if the reviewer has any additional questions.

---

> ### Author Response · Authors · 2024-11-27
>
> Dear Reviewer wHaW,
>
> We thank you again for your time and valuable feedback to help us improve our paper. After today we can no longer upload paper revisions for discussion, so we kindly ask if our responses have addressed your concerns. We look forward to hearing back from you and are happy to answer any further questions.

---

> > ### Comment · Reviewer_wHaW · 2024-11-27
> >
> > Thanks for your explanation and I don't have further questions. This is a good paper and I will keep my score.

---

### Official Review · Reviewer_WBQk · 2024-11-06

**Soundness:** 2
**Presentation:** 2
**Contribution:** 3
**Rating:** 6
**Confidence:** 4

**Summary:**

This manuscript introduces Soft Analytic Policy Optimization (SAPO), a learning framework for sequential policy optimization that combines reinforcement learning, particularly maximum entropy reinforcement learning, and differentiable simulation. The authors differentiate the soft state value function estimator and make a set of hyper-parameter and architecture modifications for a better and more stable algorithm with practical implementation. To demonstrate the effectiveness of their method, the authors implement six different simulated environments with highly heterogeneous physics, including elastic bodies, articulated bodies, and fluids. These environments are used for both locomotion and manipulation tasks with a highly efficient GPU-based implementation using Warp.

**Strengths:**

1. The paper is well-written. I did not find any mathematical mistakes or factual errors.
2. The approach of differentiating the maximum entropy reinforcement learning is valid and I personally found it interesting.
3. The results look promising with good visualizations.
4. The release of nicely implemented simulation environments can be useful for the robotics community.

**Weaknesses:**

1. The implementation details of the simulated environments are missing. How is the robotic ant simulated and actuated? What are the Young's modulus and Poisson's ratio of the elastic body? And fluid? What about the reward function for each environment?
2. I found the visual encoder part to be extremely confusing. Why a PointNet is used for visual encoder? Does the word "visual" mean point cloud instead of images here? Since the title of appendix A.1 is "Learning Visual Encoders in Differentiable Simulation", I would assume the visual encoder is learned from scratch. Why not use a pre-trained model?

Minor typo:
    Line 50: scaling -> scale

**Questions:**

1. The authors declare that the design choices have larger impact to SAPO than to SHAC: what is the reason behind it? Could the authors provide some insights?
2. I am confused by the TrajOpt baseline. I would guess that the environments are randomly initialized, which makes the trajectory optimization statistically impossible to work since it only replays the mean of learned actions. However, I found it better than some of the baselines. I might need more explanation here.
3. APG baseline also looks weird to me. Typically my experience with APG gives me impression that it instantly improves in a very short time and plateaus. However, the figures show it constantly improves. Is there a reason behind it? Or my impression is wrong?

---

> ### Author Response · Authors · 2024-11-22
> **Authors' Response for Reviewer WBQk [1/2]**
>
> We thank the reviewer for their time and positive comments on SAPO, Rewarped, and the overall presentation of our work. Based on the reviewer’s feedback, we have made changes to the paper to address each of the reviewer’s points below.
>
> - - -
>
> > [W1] The implementation details of the simulated environments are missing. How is the robotic ant simulated and actuated? What are the Young's modulus and Poisson's ratio of the elastic body? And fluid? What about the reward function for each environment?
>
> We added **two new sections** to the appendix, which should answer the reviewer’s questions. Appendix D describes the physics-based simulation techniques used in Rewarped. Appendix E describes the implementation details for every task, e.g. relevant physics hyperparameters (like Young’s modulus, Poisson’s ratio), observation/action/reward definitions.
>
> > [W2a] I found the visual encoder part to be extremely confusing. Why a PointNet is used for visual encoder? Does the word "visual" mean point cloud instead of images here?
>
> In the paper, we use “visual encoder” to refer to a network that processes some kind of visual information (e.g., point clouds, RGB/D images, LiDAR). Whereas “image encoder” or “point cloud encoder” may refer to networks which directly take images or point clouds, respectively, as inputs. As described in Appendix B.1, we subsample the particle state from simulation to use as observations. Since these observations are represented as point clouds, we use a PointNet-style network for the visual encoder.
>
> In this work, we consider SAPO with point cloud encoders only. We think SAPO should also work with image encoders, similar to other methods in model-free visual RL [[Ling et al., 2023](https://arxiv.org/abs/2306.06799)]. In that setting, using SAPO with image encoders may require a differentiable renderer in order to apply end-to-end gradient-based optimization, although differentiating through the renderer may not be necessary [[Zhang et al., 2024](https://arxiv.org/abs/2407.10648), [Luo et al., 2024](https://arxiv.org/abs/2410.03076)]. We leave this for future work, as discussed in Section 7.
>
> > [W2b] Since the title of appendix A.1 is "Learning Visual Encoders in Differentiable Simulation", I would assume the visual encoder is learned from scratch. Why not use a pre-trained model?
>
> Indeed, the visual encoder is initialized with random weights and learned from scratch. In this paper, we aim to implement the simplest version of SAPO that can solve rigid & soft body tasks. The baselines also use visual encoders with randomly initialized weights, for fair comparison. We agree with the reviewer that initializing the visual encoder with pretrained weights is a promising direction, which we leave for future work to consider.
>
> … (continue in response 2/2)

---

> ### Author Response · Authors · 2024-11-22
> **Authors' Response for Reviewer WBQk [2/2]**
>
> …
>
> > [Q1] The authors declare that the design choices have larger impact to SAPO than to SHAC: what is the reason behind it? Could the authors provide some insights?
>
> To clarify for ablation (c) in Section 6.2, which corresponds to applying design choices {III, IV, V} onto SHAC, we observe that performance improves. We find that these changes result in a 69.7% improvement over SHAC on the HandFlip task (although these changes only yield a 0.5% improvement over SHAC on the DFlex locomotion tasks, see Appendix F.3).
>
> After incorporating policy entropy regularization and soft values on top of these design choices, our full algorithm SAPO achieves 172.7% improvement over SHAC on the HandFlip task. To better understand the effects of design choices {III, IV, V}, we also added a new set of experimental results to provide individual ablations for design choices {III, IV, V}, see Appendix F.4.
>
> (Note: We updated experimental results in Section 6.1 and Section 6.2 from 6 seeds to 10 seeds, so exact values are different than the original submission. Still, we observe the same trends and improvements with SAPO.)
>
> > [Q2] I am confused by the TrajOpt baseline. I would guess that the environments are randomly initialized, which makes the trajectory optimization statistically impossible to work since it only replays the mean of learned actions. However, I found it better than some of the baselines. I might need more explanation here.
>
> Our aim for TrajOpt is to provide a simple baseline to illustrate task difficulty, that is stronger than a baseline that just takes random actions. Table 2 shows that TrajOpt outperforms model-free RL baselines on soft body tasks, while model-free RL baselines outperform TrajOpt on rigid-body tasks. This indicates that on soft-body tasks with sufficient randomization over initial state distribution $\rho_0$ and a budget of 4M/6M environment steps, model-free RL is no better than taking some mean trajectory over $\rho_0$ or even a random action baseline.
>
> > [Q3] APG baseline also looks weird to me. Typically my experience with APG gives me impression that it instantly improves in a very short time and plateaus. However, the figures show it constantly improves. Is there a reason behind it? Or my impression is wrong?
>
> In contrast to prior work [[SoftGym](https://arxiv.org/abs/2011.07215), [PlasticineLab](https://arxiv.org/abs/2104.03311), [DexDeform](https://arxiv.org/abs/2304.03223)] on soft-body manipulation with either (a) no randomization of the initial state or (b) $\rho_0$ is a discrete distribution with finite support (ie. there is some finite set of initial state configurations used across all random seeds), instead (c) $\rho_0$ is a *continuous* distribution in all our tasks. This means that performance can continue to improve over training, as new initial states can be sampled which the agent has not seen yet.
>
> Furthermore, based on our experience, what the reviewer describes may occur when using a longer horizon with APG. In this paper, for APG we do *not* use $H=T$, where $H$ is the rollout horizon and $T$ is the full episode length. Using $H=T$ (this may also be referred to as BPTT) induces a highly nonconvex optimization landscape that can cause methods to get stuck in local optima [[Xu et al., 2022](https://arxiv.org/abs/2204.07137)]. Instead, we use the same rollout horizon $H=32$ across all baselines for fair comparison. Combining APG with the short-horizon idea (from SHAC and, more broadly, truncated BPTT) has also been shown to work better on tasks in Brax as well (see [here](https://github.com/google/brax/pull/476)).
>
> However, we have found that SHAC can still get stuck in local optima, see Appendix F.1. In this paper, we propose to mitigate these issues with maximum entropy RL and thus formulate SAPO.
>
> - - -
>
> **Overall**, we believe the changes should resolve all of the reviewer’s concerns, and hope the reviewer will increase their scores accordingly. If there are additional questions, we would be happy to discuss them further, please let us know!

---

> ### Author Response · Authors · 2024-11-27
>
> Dear Reviewer WBQk,
>
> Thank you again for your time and valuable feedback to help us improve our paper. After today we can no longer upload paper revisions for discussion, so we kindly ask if our rebuttal has addressed your concerns. Thus far, two out of four reviewers have engaged with us and updated their scores during the discussion period.
>
> We look forward to hearing back from you and are happy to answer any further questions.

---

> > ### Comment · Reviewer_WBQk · 2024-11-28
> >
> > I appreciate the time and effort that went into the rebuttal. The authors’ responses resolved my concern about TrajOpt and APG. The implementation details of the environments are also helpful to the community. I will raise my score to 6 to reflect this. I believe this work is valuable to the differentiable simulation and learning audiences and would appreciate if the authors could open-source the work after the acceptance.

---

> ### Author Response · Authors · 2024-11-28
>
> Dear Reviewer WBQk,
>
> Thank you for considering our responses, and we are happy to hear those concerns have been resolved. Per the reproducibility statement in the paper, we remain committed to releasing all code after publication for the benefit of the community. We are encouraged to hear that the reviewer believes this work will be valuable to the differentiable simulation and learning communities.
>
> **Overall**, if you have any remaining questions or concerns that stand between us and a higher score, we are happy to answer further questions given the extended discussion period.
>
> ---
>
> **Edit**: As [W1, Q2, Q3] were resolved according to the reviewer's last reply, if there are still lingering concerns on the remaining two points: [see below].

---

> ### Author Response · Authors · 2024-11-29
>
> Dear Reviewer WBQk,
>
> We are thankful for your time and valuable feedback to help us improve our paper. We are encouraged by your comments that our paper is "well-written" with "good visualizations", our approach is "valid" and "interesting",  and that our work is "valuable to the differentiable simulation and learning audiences".
>
> Given the remaining discussion period and the reviewer's latest reply, we would greatly appreciate if the reviewer could please clarify any remaining concerns they have that stand between us and a higher score. If anything was missed, we aim to resolve them and further improve our work.
>
> ---
>
> > ... "would appreciate if the authors could open-source the work after the acceptance."
>
> As stated in the reproducibility statement in the paper, we remain committed to releasing of all code, including algorithms and simulation tasks, upon publication.
>
> > [W2]
>
> We edited the phrasing in Appendix B.1 regarding visual encoders. We also added a **new Figure 4** in the appendix (on page 17), which illustrates the computational graph of SAPO (including the encoders). If the reviewer believes their concerns with [W2] were not resolved, we are happy to take suggestions to further improve clarity.
>
> > [Q1]
>
> To reiterate, in Section 6.2 we *do not declare* that the design choices have larger impact to SAPO than to SHAC. With ablation (c), we observe that SHAC also benefits from design choices {III, IV, V}. In Appendix F.4, we provide additional ablations to show that each design choice benefits SAPO. From Table III, we find that SAPO outperforms SHAC. This is because SAPO benefits from its maximum entropy RL formulation using policy entropy regularization and soft values, in contrast to SHAC. Section 4.2 provides insight for why we use design choices {III, IV, V}. Design choices {I, II} are specific to maximum entropy RL, and thus do not apply to SHAC.
>
> ---
>
> **Overall**, we believe this should further resolve any of the reviewer's remaining concerns, and hope the reviewer will increase their scores accordingly. We are happy to discuss additional questions given the extended discussion period.

---

### Author Response · Authors · 2024-11-22
**Overview Response for Discussion Phase**

We thank the reviewers for their time and feedback on our work. We are encouraged to see that all reviewers acknowledge both our contribution on parallel differentiable simulation (Rewarped) and our contribution on first-order model-based RL (SAPO), as strengths of our work.

We briefly summarize the changes we’ve made since the initial submission for the discussion period, taking into consideration all the reviewers’ feedback.

- - -

- New section (Appendix A) that extends the related work section, to additionally cover model-based RL and to include prior work on *non-parallel* differentiable simulation that did not fit in the main body of the paper. **[cLkG, W9i4]**

- New section (Appendix D) that reviews the underlying physics-based simulation techniques used in Rewarped. **[WBQk, cLkG]**

- New section (Appendix E) with details on Rewarped tasks, including observation space, action space, reward definitions, physics settings, etc. **[WBQk, cLkG]**

- New subsection (Appendix F.5) with results on the runtime performance and scalability of Rewarped. **[W9i4, wHaW]**

- Expanded ablation studies (Appendix F.4) with new results for individual ablations on design choices {III, IV, V}. **[wHaW]**

- Updated experimental results for Rewarped tasks with 10 random seeds. (We observe the same trends as with the results in the initial submission with 6 random seeds, so our analysis is the same.)

- - -

We also respond to the reviewers individually to address their feedback. We believe these changes should resolve all of the reviewers’ notes, and hope the reviewers will consider raising their scores accordingly.

---

### Meta-Review · Area_Chair_xatb · 2024-12-21

**Metareview:**

The paper proposes a new algorithm that combines max-entropy RL and differentiable physics simulators. All reviewers feel very positive about the paper. The paper is clearly written, well organized and presenting strong results. The paper gives good background information, motivation. The proposed design mitigates the complexity of offline policy gradient evaluation while still benefiting from the simulator’s analytical gradients, and the paper gives informative design decisions for practitioners too. In addition to the algorithm, the paper presents a new differentiable simulator of soft body dynamics with good visualization. which would be beneficial to the robotics community going forward. Overall, this is a strong, well-written paper with applauding contributions in both algorithm and simulator design.

**Additional Comments On Reviewer Discussion:**

Reviewer WBQk raises questions about missing details, and writing clarity; reviewer wHaW raises concerns on novelty, generality, and missing details; Reviewer W9i4 raises concerns on novelty; Reviewer cLkG raises questions about missing details, and missing related work. All these are addressed in the discussion.

---

### Decision · Program_Chairs · 2025-01-22

Accept (Spotlight)